# SUBMODULAR FUNCTION MINIMIZATION WITH DUELING ORACLE

**Kaien Sho[1]**    **Shinji Ito[1,2]**
[1]The University of Tokyo    [2]RIKEN
sho-kaien1023@g.ecc.u-tokyo.ac.jp, shinji@mist.i.u-tokyo.ac.jp

## ABSTRACT

We consider submodular function minimization using a *dueling oracle*, a noisy pairwise comparison oracle that provides relative feedback on function values between two queried sets. The oracle's responses are governed by a *transfer function*, which characterizes the relationship between differences in function values and the parameters of the response distribution. For a *linear* transfer function, we propose an algorithm that achieves an error rate of $O(n^{\frac{3}{2}}/\sqrt{T})$, where $n$ is the size of the ground set and $T$ denotes the number of oracle calls. We establish a lower bound: Under the constraint that differences between queried sets are bounded by a constant, any algorithm incurs an error of at least $\Omega(n^{\frac{3}{2}}/\sqrt{T})$. Without such a constraint, the lower bound becomes $\Omega(n/\sqrt{T})$. These results show that our algorithm is optimal up to constant factors for constrained algorithms. For a *sigmoid* transfer function, we design an algorithm with an error rate of $O(n^{\frac{7}{5}}/T^{\frac{2}{5}})$, and establish lower bounds analogous to the linear case.

## 1 INTRODUCTION

Let $f$ be a set function defined on subsets of a finite set $[n] = \{1, \cdots, n\}$. A function $f$ is called *submodular* if it satisfies $f(X) + f(Y) \geq f(X \cup Y) + f(X \cap Y)$ for all $X, Y \subseteq [n]$.

Submodular functions are closely related to convex functions (Lovász, 1982; Fujishige, 1991; Bach, 2011) and play a significant role in numerous problems. Thus, *submodular function minimization* (SFM) arises in a wide range of research fields, including machine learning (Bilmes, 2022), operations research (Hochbaum & Hong, 1995; Queyranne & Schulz, 1995), combinatorial optimization (Lovász, 1982; Edmonds, 2001; Schrijver, 2003), game theory (Shapley, 1971; Topkis, 1998), and economics (Stigler & Samuelson, 1948; Topkis, 1998; Vives, 1999). In machine learning and artificial intelligence, SFM has found use in tasks such as graphical models (Kolmogorov & Zabih, 2002; Krause & Guestrin, 2005), PAC-learning (Narasimhan & Bilmes, 2004), clustering (Narasimhan et al., 2005; Narasimhan & Bilmes, 2007), and image segmentation (Kohli & Torr, 2010; Jegelka & Bilmes, 2011).

In optimization problems, many studies assume access to first- or zero-order oracles that provide gradients or exact function values at queried points. However, in real-world scenarios, it is often impractical or unreliable to obtain precise gradients or function values. Instead, feedback based on relative evaluations, which determines which of two options is preferable, tends to be more feasible, efficient, and robust.

For example, in machine learning, data collection through relative evaluations not only improves reliability but also facilitates the acquisition of larger datasets. Pairwise comparison queries are widely used in applications such as feedback collection for large language models (LLMs) (Liusie et al., 2023; Wu et al., 2023; Jiang et al., 2023; Liu et al., 2024) and reinforcement learning with human feedback (RLHF) (Sadigh et al., 2017; Pacchiano et al., 2021; Azar et al., 2024; Zhu et al., 2023).

Research on optimization problems using a dueling oracle has been limited to multi-armed bandit problems (Yue et al., 2012; Dudík et al., 2015; Sui et al., 2018; Saha et al., 2021b; Saha & Gaillard, 2022) and convex optimization (Saha et al., 2021a; 2025; Blum et al., 2024). In the context of

Table 1: Upper and lower bound for submodular function minimization with a dueling oracle.

| Transfer Function | Linear | Sigmoid | General |
|---|---|---|---|
| Upper Bound | $O\left(\dfrac{n^{\frac{3}{2}}}{\sqrt{T}}\right)$ | $O\left(\dfrac{n^{\frac{7}{5}}}{T^{\frac{2}{5}}}\right)$ | $O\left(\dfrac{n^{\frac{4}{3}}}{T^{\frac{1}{3}}}\right)$ |
| Lower Bound (with Restriction 1) | $\Omega\left(\dfrac{n^{\frac{3}{2}}}{\sqrt{T}}\right)$ | $\Omega\left(\dfrac{n^{\frac{3}{2}}}{\sqrt{T}}\right)$ | $\Omega\left(\dfrac{n^{\frac{3}{2}}}{\sqrt{T}}\right)$ |
| Lower Bound (without Restriction 1) | $\Omega\left(\dfrac{n}{\sqrt{T}}\right)$ | $\Omega\left(\dfrac{n}{\sqrt{T}}\right)$ | $\Omega\left(\dfrac{n}{\sqrt{T}}\right)$ |

Submodular Function Minimization (SFM), the setting in which only a dueling oracle is used without access to a value oracle has not been addressed so far, making this study the first to explore it.

Existing studies on SFM assume access to an oracle that provides the exact or noisy function value for a queried set. The first polynomial-time algorithm for this problem was introduced by Grötschel et al. (1981), employing the ellipsoid method. Combinatorial strongly polynomial algorithms were developed by Iwata et al. (2000) and Schrijver (2003). In parallel, Hazan & Kale (2012) explored online optimization for submodular functions, focusing on iterative decision making and evaluating performance through regret, defined as cumulative error during iterations. In addition, Ito (2019) investigated SFM in the context of noisy function value oracles.

In this paper, we consider SFM with a dueling oracle. A dueling oracle, or a noisy pairwise comparison oracle, provides probabilistic binary feedback indicating which of two queried sets has a higher function value, without disclosing the actual function values. The algorithm can query a pair of sets, and the dueling oracle only returns a probabilistic response in $+1$ or $-1$, indicating which set has the higher function value. The probability of receiving the correct response increases as the difference in function values grows, while smaller differences result in outcomes that are nearly evenly distributed. This reflects the concept of a *duel*, where larger differences favor the stronger contender, while smaller differences introduce more uncertainty in response. This problem setting is motivated by applications such as recommendation, as illustrated in Examples 1 and 2 in Section 3. The relationship between differences in function values and parameters of the response distribution is characterized by a function $\rho : [-1, 1] \to [-1, 1]$, called a *transfer function*.

Our contribution is twofold. First, we propose efficient algorithms for SFM using a dueling oracle with linear, sigmoid, and general nonlinear transfer functions. The reason for focusing on these two concrete examples is explained in Section 3. Second, we establish lower bounds for each setting. As shown in Table 1, for the case of a linear transfer function, we develop an algorithm that achieves an error bound of $O(n^{\frac{3}{2}}/\sqrt{T})$ for any submodular function. For sigmoid transfer functions, we propose an algorithm with an upper bound of $O(n^{\frac{7}{5}}/T^{\frac{2}{5}})$, and for general nonlinear transfer functions, we provide an algorithm with an upper bound of $O(n^{\frac{4}{3}}/T^{\frac{1}{3}})$. Regarding lower bounds, we show that any algorithm satisfying the constraint that the element-wise difference between the two sets queried by the dueling oracle is of constant order must incur an error of at least $\Omega(n^{\frac{3}{2}}/\sqrt{T})$ in a hard instance. Furthermore, we prove that any algorithm, without restrictions, must suffer an error of at least $\Omega(n/\sqrt{T})$. This result implies that, for the linear transfer function, our proposed algorithm is optimal among those satisfying the given constraint, and even with unrestricted algorithms, the error cannot be improved by more than a factor of $O(1/\sqrt{n})$. We here note that, while the formulas in Table 1 represent the error achievable with a given number of queries $T$, these can also be interpreted as an evaluation of the number of queries required to achieve an error below a given threshold $\epsilon > 0$. For example, the upper bound of $O(n^{\frac{3}{2}}/\sqrt{T})$ in the table indicates that an error no greater than $\epsilon$ can be attained with $T \propto n^3/\epsilon^2$ queries.

Our algorithm leverages the Lovász extension (Lovász, 1982), which extends a submodular function to a continuous convex function, and employs the stochastic gradient descent (SGD) method for convex minimization. A key property of the Lovász extension is that the minimum of the original submodular function coincides with the minimum of the extended convex function. Thus, solving the convex minimization problem via SGD yields a solution to SFM. The update direction in SGD is determined by an estimator of a subgradient of the convex function. To minimize the error introduced by SGD, reducing the bias and variance of this estimator is crucial. In particular, an unbiased

estimator, whose expectation matches the true subgradient, leads to accurate solutions. The approach using the Lovász extension and SGD has already been explored by Hazan & Kale (2012); Ito (2019; 2022). However, unlike these studies, direct access to function values is unavailable in our setting. While these works construct unbiased estimators of subgradients using the responses from a function value oracle, we estimate subgradients using responses from a dueling oracle.

For the linear transfer function, unbiased estimators of subgradients can be constructed by using responses from a dueling oracle. The general flow of the algorithm is similar to those proposed by Hazan & Kale (2012); Ito (2019), the difference being in the construction of the unbiased estimator.

In contrast, for the sigmoid transfer function, the construction of unbiased subgradient estimators becomes infeasible, which poses a significant challenge. Using biased estimators in SGD can cause error accumulation across iterations, resulting in considerable inaccuracies. In related work, Saha et al. (2025) addressed a similar issue in convex optimization with a dueling oracle. They considered a gradient descent-based algorithm under similar conditions, where unbiased subgradient estimators were unavailable. Their approach relies on the assumptions of $\beta$-smoothness and $\alpha$-strong convexity of the objective function to control errors. Unfortunately, the convex function derived from the Lovász extension of a general submodular function does not satisfy these properties, making Saha et al. (2025)'s error control techniques inapplicable.

To overcome this difficulty, we incorporate Firth's method (Firth, 1993) to reduce the bias in maximum likelihood estimators (MLE). Firth's method significantly reduces bias in estimation problems related to the natural parameters of the exponential family of distributions. In this study, when the transfer function of the dueling oracle is sigmoid, the resulting model corresponds to a logistic regression, making it possible to apply Firth's method. By incorporating Firth's method into SGD, we mitigate the impact of accumulated bias, reducing the total error, and enabling an effective optimization algorithm. To our knowledge, no existing SFM algorithm based on SGD avoids the use of unbiased subgradient estimators derived from the Lovász extension. This is the significance of this work.

The inability to use an unbiased estimator makes not only the algorithm design but also its analysis and parameter tuning considerably more challenging. In existing studies (Hazan & Kale, 2012; Ito, 2019), the analysis of the optimization error typically focused on two terms that increase or decrease with the learning rate $\eta$, and choosing $\eta$ to balance these two terms yields the desired error (or regret) bound. In contrast, in the present setting, a third term arises from the bias of the gradient estimator, and an additional parameter, the number of oracle calls $k$ used for gradient estimation, affects these terms, resulting in a more complex analysis. In this work, we provide a new parameter-setting rule for $\eta$ and $k$ that balances all three terms so that the overall error bound is minimized.

The proof of the lower bound is based on the techniques of Ito (2019) and Auer et al. (2002). Ito (2019) establishes lower bounds for SFM, while Auer et al. (2002) addresses a lower bound for a bandit problem. While the construction of the objective function follows prior work (Ito, 2019), the lower-bound analysis presents a significant challenge, as the techniques used in previous studies cannot be directly applied. This difficulty arises because the oracle and feedback structure assumed in existing models differ substantially from those considered in this work. Intuitively, prior work allows queries on a single subset, whereas our setting permits queries on *pairs* of subsets. Since the number of feasible choices for data collection is therefore much larger in our framework, proving lower bounds becomes considerably more nontrivial. In this paper, we address this challenge by extending the analysis to incorporate the probability distribution over queried pairs and by analyzing the distance between the resulting distributions.

Leveraging Yao's principle, we bound the error of randomized algorithms by the error of deterministic algorithms under a carefully chosen distribution. For deterministic algorithms, the KL divergence of the algorithm's output is bounded by the KL divergence across all input sequences. Using these results, we demonstrate that for two objective functions, the KL divergence of their respective input sequences, and consequently, the KL divergence of their output, remains small. Finally, we show that for two objective functions with different optimal solutions, any algorithm can only exhibit limited changes in its output and thus incurs a non-negligible error.

## 1.1 RELATED WORK

**Online submodular minimization in the bandit setting**

The online submodular minimization introduced by Hazan & Kale (2012) is closely related to our algorithm. In this setting, an online decision maker iteratively selects a subset $S_t \subseteq [n]$ over $t = 1, \cdots, T$. After each selection, the decision maker incurs a loss of $f_t(S_t)$, where each $f_t$ is a submodular function. The performance of the algorithm is evaluated by the *regret* defined as $\text{Regret}_T := \sum_{t=1}^{T} f_t(S_t) - \min_{S \subseteq [n]} \sum_{t=1}^{T} f_t(S)$.

In the bandit setting, only the loss $f_t(S_t)$ for the chosen subset $S_t$ is observed, and no other information is available. The algorithm proposed by Hazan & Kale (2012) is based on the Online Gradient Descent algorithm of Zinkevich (2003), using unbiased subgradient estimators derived from the Lovász extension extension of the submodular functions. The key idea of this algorithm is to dynamically adjust the probabilities of choosing a subset and use just one sample for both exploration and exploitation. The regret achieved by this algorithm is $O\left(nT^{\frac{2}{3}}\right)$, which translates to an error of $O\left(n/T^{\frac{1}{3}}\right)$ when applied directly to SFM.

**Submodular function minimization with noisy oracles**

Our work is also influenced by the study of SFM using a noisy zero-order oracle by Ito (2019); Lattimore (2024). In their work, at each step $t = 1, \cdots, T$, a subset $S_t$ is chosen, and a noisy observation $\hat{f}_t(S_t)$ of the submodular objective function $f$ is obtained, where $\hat{f}$ satisfies $\mathbb{E}[\hat{f}_t(S)] = f(S)$ for all $S \subseteq [n]$ and $t = 1, \cdots, T$.

The algorithm in their research is also based on SGD, with the final solution obtained as the average of the iterations over $T$ steps. Their approach achieves an error of $O\left(n^{\frac{3}{2}}/\sqrt{T}\right)$, which improves to $O\left(n/\sqrt{T}\right)$ under the assumption that each noisy observation $\hat{f}_t$ is also submodular.

Furthermore, their approach also influences the proof of the lower bound for SFM in our study. They constructed a distribution of objective functions that serves as a hard instance, showing that even when a distribution is perturbed, the observed values change slightly, making it challenging for any algorithm to distinguish between distributions. This was demonstrated using KL divergences of distributions.

**Convex optimization with dueling oracle**

Convex optimization using dueling oracles has been explored in Saha et al. (2021a) and Saha et al. (2025), which consider transfer functions $\rho$ with specific properties. In Saha et al. (2021a), $\rho$ is assumed to be a sign function, while Saha et al. (2025) considers a function that satisfies $\rho'(x) \geq c_\rho p |x|^{p-1}$ for all $x \in (-r, r)$, where $p \geq 1$, $r > 0$, and $c_\rho > 0$. Both studies propose Gradient Descent-based algorithms; however, unbiased estimators of subgradients cannot be constructed from the information of dueling oracles.

To address this difficulty, Saha et al. (2021a) determined the descent direction using the normalized subgradient $\nabla f(x)/\|\nabla f(x)\|$, and demonstrates that the algorithm produces an approximate solution. Meanwhile, Saha et al. (2025) introduces a scaled gradient approach using $\nabla f(x)\|\nabla f(x)\|^{p-1}$ and shows that an approximate solution exists within the sequence of points generated by the algorithm. Both approaches rely on the assumptions of $\beta$-smoothness and $\alpha$-strong convexity of the objective function, which enable the analysis of convergence and approximation guarantees.

## 2 PRELIMINARIES

### 2.1 SUBMODULAR FUNCTIONS

Let $n$ be a positive integer and let $[n] = \{1, 2, \ldots, n\}$. Denote by $2^{[n]}$ the power set of $[n]$, i.e. the set of all subsets of $[n]$. We consider a set function $f : 2^{[n]} \to [0, 1]$ defined in the decision space $2^{[n]}$.

A function $f : 2^{[n]} \to [0, 1]$ is submodular if and only if for all sets $X, Y \in 2^{[n]}$ such that $X \subseteq Y$ and for all elements $i \in [n] \setminus Y$, we have:

$$f(X \cup \{i\}) - f(X) \geq f(Y \cup \{i\}) - f(Y). \tag{1}$$

## 2.2 LOVÁSZ EXTENSION

The Lovász extension is a fundamental technique in designing algorithms for submodular function minimization. Although a submodular function $f$ is originally defined in the decision space $2^{[n]}$, it can be regarded as the vertices of the hypercube $\mathcal{K} = [0, 1]^n$. The Lovász extension $\hat{f}$ provides a continuous extension of a submodular function $f$ to the whole interior of $\mathcal{K}$.

The Lovász extension is constructed by dividing the hypercube $\mathcal{K}$ into $n!$ regions and defining $\hat{f}$ as a piecewise linear function that interpolates between the values of the function at the vertices of $\mathcal{K}$.

**Definition 1.** Given a function $f : 2^{[n]} \to [0, 1]$, the Lovász extension $\hat{f} : \mathcal{K} \to [0, 1]$ is defined as follows; for $w \in \mathcal{K}$, order the components in decreasing order $w_{\pi(1)} \geq \cdots \geq w_{\pi(n)}$, where $\pi : [n] \to [n]$ is a permutation. Let $w_{\pi(0)} = 1$, $B_i = \{\pi(1), \ldots, \pi(i)\}$ for $i \in [n]$ and $B_0 = \varnothing$. The value of the Lovász extension $\hat{f}(w)$ is defined as:

$$\hat{f}(w) = \sum_{i=1}^{n} w_{\pi(i)}[f(B_i) - f(B_{i-1})] + f(\varnothing) = \sum_{i=0}^{n-1} f(B_i)(w_{\pi(i)} - w_{\pi(i+1)}) + f([n]). \quad (2)$$

An alternative equivalent expression for the Lovász extension is also utilized to construct our algorithms.

**Proposition 1.** Let $\hat{f}$ be a Lovász extension of a function $f : 2^{[n]} \to [0, 1]$. For any $w \in \mathcal{K}$, the following equations hold:

$$\hat{f}(w) = \int_0^1 f(\{i \mid w_i \geq z\})\mathrm{d}z = \mathbb{E}_{z \sim \mathrm{Unif}([0,1])}[f(\{i \mid w_i \geq z\})]. \quad (3)$$

The following theorem provides a connection between submodular function minimization and convex function minimization.

**Theorem 1.** *(Fujishige (1991)) A function $f : 2^{[n]} \to [0, 1]$ is submodular if and only if its Lovász extension $\hat{f} : \mathcal{K} \to [0, 1]$ is convex. For a submodular function $f : 2^{[n]} \to [0, 1]$ and its Lovász extension $\hat{f} : \mathcal{K} \to [0, 1]$, we have $\min_{X \in 2^{[n]}} f(X) = \min_{w \in \{0,1\}^n} \hat{f}(w) = \min_{w \in [0,1]^n} \hat{f}(w)$.*

When analyzing the minimization of the convex function $\hat{f}$, it is notable that $\hat{f}$ is piecewise linear. Therefore, its subgradient is constant within each region of linearity, as formally stated in the following proposition.

**Proposition 2.** *Let $f$ be a submodular function. For $w \in \mathcal{K}$, let $\pi$ be a permutation that orders the components of $w$ in decreasing order, and let $\tau$ be the inverse permutation of $\pi$. Then, a subgradient $g$ of $\hat{f}$ at $w$ is given as follows:*

$$g_{\pi(i)} = f(B_i) - f(B_{i-1}) \quad \text{or, equivalently,} \quad g_i = f(B_{\tau(i)}) - f(B_{\tau(i)-1}).$$

## 2.3 STOCHASTIC GRADIENT DESCENT

By Theorem 1, submodular minimization can be reformulated as convex minimization. Therefore, our algorithm is based on the *stochastic gradient descent* (SGD) method for convex minimization.

The algorithm initializes $w^{(1)} = \frac{1}{2} \cdot \mathbf{1} \in \mathcal{K}$. For each iteration $t = 1, 2, \ldots, T$, the point $w^{(t)}$ is updated to $w^{(t+1)}$ using information about the objective function $\hat{f}$ obtained through oracle queries. At each update, an estimator $\hat{g}_t$ of a subgradient of $\hat{f}$ at $w^{(t)}$ is constructed, and the update rule is given by: $w^{(t+1)} = \Pi_{\mathcal{K}}(w^{(t)} - \eta\hat{g}_t)$, where $\eta > 0$ is a learning rate parameter that can be arbitrarily chosen. Here, $\Pi_{\mathcal{K}} : \mathbb{R}^n \to \mathcal{K}$ represents the Euclidean projection onto $\mathcal{K}$, defined as: $\Pi_{\mathcal{K}}(w) = \arg\min_{v \in \mathcal{K}} \|v - w\|_2$. Since $\mathcal{K} = [0, 1]^n$, the projection is computationally efficient. For any $v \in \mathbb{R}^n$, the projection $w = \Pi_{\mathcal{K}}(v)$ can be implemented by clipping each component of $v$ to the interval $[0, 1]$: $w_i = \min\{1, \max\{0, v_i\}\}$.

For the sequence of points $\{w^{(t)}\}_{t=1}^{T}$ generated by this procedure, the average point $\bar{w} = \frac{1}{T}\sum_{t=1}^{T} w^{(t)}$ satisfies the following error bound, as stated in Hazan & Kale (2012, Lemma 11).

**Theorem 2.** *Let* $\hat{f} : \mathcal{K} \rightarrow [0, 1]$ *be a convex function in the hypercube* $\mathcal{K} = [0, 1]^n$. *Let* $w^{(1)}, w^{(2)}, \ldots, w^{(T)}$ *be defined by* $w^{(1)} = \frac{1}{2} \cdot \mathbf{1}$ *and* $w^{(t+1)} = \Pi_{\mathcal{K}}(w^{(t)} - \eta \hat{g}_t)$. *When* $\hat{g}_1, \hat{g}_2, \ldots, \hat{g}_T$ *are unbiased estimators of subgradients, i.e.,* $\mathbb{E}[\hat{g}_t \mid w^{(t)}] = g_t$, *where* $g_t$ *is a subgradient of* $\hat{f}$ *at* $w^{(t)}$, *then* $\bar{w} := \frac{1}{T} \sum_{t=1}^{T} w^{(t)}$ *satisfies* $\mathbb{E}[\hat{f}(\bar{w})] - \min_{w^* \in \mathcal{K}} \hat{f}(w^*) \leq \frac{1}{T} \left( \frac{n}{8\eta} + \frac{\eta}{2} \sum_{t=1}^{T} \mathbb{E}[\|\hat{g}_t\|_2^2] \right)$.

## 3 PROBLEM STATEMENT

We address the problem of minimizing a submodular function $f : 2^{[n]} \rightarrow [0, 1]$. In this setting, the exact values of $f$ are not directly accessible; instead, we rely on a dueling oracle, in other words, a noisy comparison oracle of two points. The dueling oracle provides a random binary response $o \in \{\pm 1\}$ for a pair of subsets $(S, S') \in 2^{[n]} \times 2^{[n]}$. Note that, as in prior works on dueling bandits and dueling convex optimization (Saha et al., 2021b;a), all outputs from the dueling oracle are assumed to be independent. Therefore, even if the same pair is queried multiple times, the outputs are not necessarily identical. The probability of the response is given as: $\Pr(o = +1) = \frac{1}{2} + \frac{1}{2}\rho(f(S) - f(S'))$, $\Pr(o = -1) = \frac{1}{2} - \frac{1}{2}\rho(f(S) - f(S'))$, where $\rho : [-1, 1] \rightarrow [-1, 1]$ is a fixed *transfer function* that maps the difference in function values to the distribution parameter.

Our goal is to design algorithms that minimize additive error $E_T := f(\hat{S}) - \min_{S \in 2^{[n]}} f(S)$, where $\hat{S}$ is the algorithm output. The algorithm is given the decision set $2^{[n]}$, the number of available oracle calls $T$, and the transfer function $\rho$. At each iteration $t = 1, 2, \ldots, T$, the algorithm chooses a pair of subsets $(S_t, S_t')$ and observes a response $o_t \in \{\pm 1\}$ from the dueling oracle. After $T$ iterations, the algorithm outputs a subset $\hat{S} \in 2^{[n]}$.

This problem is motivated by the following applications:

**Example 1** (recommendation system). Let us consider recommendation systems, in which we want to find a collection of items $S \subseteq [n]$ that maximizes a user's satisfaction $f(S)$, based on feedback obtained after presenting $S$ to various users. In practice, users rarely provide reliable cardinal scores but can often express pairwise preferences between presented collections; such comparative feedback is naturally modeled by a dueling oracle. Under certain conditions the resulting optimization can be cast as submodular minimization. Let $x \in \{0, 1\}^n$ be the indicator of $S$ and suppose user utility $f(S)$ has the quadratic form as $f(S) = \sum_{i=1}^{n} a_i x_i + \sum_{1 \leq i \leq j \leq n} b_{ij} x_i x_j$. If $b_{ij} \geq 0$ for all $i, j$, which models complementary interactions like camera and lens, then $f$ is supermodular (Nemhauser et al., 1978; Boros & Hammer, 2002). Hence, maximizing $f$ is equivalent to minimizing $-f$, which is submodular, and thus the problem is covered by our dueling-oracle SFM framework. Note that the quadratic form above is only illustrative; our approach applies to general supermodular maximization problems. More broadly, any selection task defined on a ground set, such as choosing keywords or response components for a chatbot, can be addressed by the same framework.

**Example 2** (multi-product price optimization). Consider price optimization over a set of products to maximize total revenue. As shown in (Ito & Fujimaki, 2016), the total revenue function can be formulated as a supermodular function under suitable assumptions, which means that the problem is an instance of submodular minimization. In this setting, relative evaluations—such as pairwise comparisons between pricing vectors—are often less sensitive to external factors like weather or seasonality than absolute measurements, and thus more accurately capture the effect of price changes.

Following prior work on dueling convex optimization (Saha et al., 2021a; 2025), we consider a transfer function $\rho$ that satisfies the following properties:

1. Strictly monotonically increasing function: $\rho(x) > \rho(y)$ if $x > y$.
2. Odd function: $\rho(-x) = -\rho(x)$ for all $x \in [-1, 1]$. In particular, $\rho(0) = 0$.

In particular, this study focuses on the following two examples of $\rho$:

- **Sigmoid transfer function**: $\rho(x) = \frac{2}{1 + e^{-bx}} - 1, \quad b > 0$.
- **Linear transfer function**: $\rho(x) = ax, \quad 0 < a \leq 1$.

Our primary motivation for considering the sigmoid transfer function is that it is both natural and important, particularly due to its connection with the Bradley–Terry model (Bradley & Terry, 1952).

The Bradley–Terry model has long been used as one of the most standard models for pairwise comparisons, and its ability to fit a wide range of empirical data has been well documented (David, 1988; Cattelan, 2012). Its practical usefulness has been demonstrated across diverse applications, including ranking systems, search engine click modeling (Joachims, 2002), preference estimation in recommender and information-retrieval systems (Chen et al., 2013), and preference modeling for post-training large language models (Stiennon et al., 2020; Rafailov et al., 2023). This models the probability that item $i$ is preferred over item $j$ as $\Pr(i \succ j) = \frac{\exp(\beta_i)}{\exp(\beta_i) + \exp(\beta_j)}$, where $\beta_i, \beta_j \in \mathbb{R}$ are parameters associated with the items. Now, suppose that for each subset $S$, the corresponding parameter $\beta_S$ is proportional to the utility $-f(S)$, i.e., $\beta_S = -bf(S)$ for some $b > 0$. In this case, the output of a dueling oracle based on a sigmoid transfer function is consistent with the Bradley–Terry model as $\Pr(o = +1) = \frac{1}{2} + \frac{1}{2}\rho(f(S') - f(S)) = \frac{1}{1 + \exp(-b(f(S) - f(S')))} = \frac{\exp(\beta_{S'})}{\exp(\beta_{S'}) + \exp(\beta_S)}$. This model has been widely used as a standard tool for modeling preferences based on pairwise comparisons, and appears in various domains such as sports ranking, consumer behavior, and AI-based decision-making. Moreover, the Bradley–Terry model can also be interpreted probabilistically—for example, as the probability that a random variable drawn from an exponential distribution associated with item $i$ exceeds that of item $j$. This interpretation further supports its relevance as a model for real-world phenomena.

The linear transfer function is important for two main reasons. First, any smooth transfer function can be approximated by a linear function in a small neighborhood around zero. Therefore, even if the true underlying transfer function is nonlinear, the linear case becomes relevant when the exploration is restricted to regions where the difference in values is small. In such settings, approaches developed for the linear case are expected to remain effective. Numerical results supporting this hypothesis to some extent are provided in Section B.2 of the appendix. Second, the linear case is the simplest from the perspective of algorithm design and analysis. It thus serves as a natural starting point for investigating our new problem setting.

## 4 LINEAR TRANSFER FUNCTION

In this section, we present an algorithm for submodular minimization using a dueling oracle with a linear transfer function. The proposed algorithm is based on the Stochastic Gradient Descent on the Lovász extension $\hat{f}$ of the submodular function $f$. It achieves an additive error bound of $O(\frac{n^{\frac{3}{2}}}{a\sqrt{T}})$. Furthermore, we prove that, under the condition that the difference between the elements in the two queried sets is bounded by a constant order, any algorithm suffers an error of at least $\Omega(\frac{n^{\frac{3}{2}}}{a\sqrt{T}})$. When there are no restrictions on queries, the lower bound remains $\Omega(\frac{n}{a\sqrt{T}})$.

### 4.1 ALGORITHM

Our algorithm is based on the SGD method. To construct subgradient estimators in SGD, we use information using a dueling oracle. When the transfer function $\rho$ of the dueling oracle is linear, we can construct unbiased estimators of subgradients by the definition of the dueling oracle. From Proposition 2, querying $(B_{\tau(i)}, B_{\tau(i)-1})$ provides $o_t$ that corresponds to the $i$-th component of the subgradient $g$. By the linearity of expectation, we have $\mathbb{E}[\frac{o_t}{a}] = f(B_{\tau(i)}) - f(B_{\tau(i)-1}) = g_i$.

From Proposition 1, the optimal value of the submodular function $f$ over $2^{[n]}$ is equivalent to the optimal value of its Lovász extension $\hat{f}$ over $\mathcal{K}$. Using this equivalence, the algorithm performs SGD to find a solution $\bar{w} \in \mathcal{K}$ with a small error relative to the optimal solution of $\hat{f}$. Finally, leveraging the representation of the Lovász extension in equation 3, the algorithm outputs a set $\hat{S}_T$ such that $\mathbb{E}[f(\hat{S}_T)] = \hat{f}(\bar{w})$.

This algorithm is a simple extension of the algorithms proposed by Hazan & Kale (2012) and Ito (2019). In these prior works, the algorithms rely on querying sets to obtain function values, which are then used to construct unbiased estimators of the subgradients. In contrast, this algorithm leverages the properties of the dueling oracle to perform SGD without directly using function values.

---

**Algorithm 1** Submodular Stochastic Gradient Descent for linear transfer function

---

**Input:** The size $n \geq 1$ of the ground set, the number $T \geq 1$ of oracle calls and the coefficient $0 < a \leq 1$ of the transfer function $\rho(x) = ax$.

1: Set the initial point $w^{(1)} = \frac{1}{2} \cdot \mathbf{1}$, the learning rate $\eta = \frac{a}{2\sqrt{nT}}$ and the number of steps $T' = \frac{T}{n}$.
2: **for** $t = 1, 2, \ldots, T'$ **do**
3:     Find a permutation $\pi$ corresponding to $w^{(t)}$, i.e., $w^{(t)}_{\pi(1)} \geq \cdots \geq w^{(t)}_{\pi(n)}$, and its inverse permutation $\tau$.
4:     Define $B_i = \{\pi(1), \ldots, \pi(i)\}$ for $i \in [n]$ and $B_0 = \varnothing$.
5:     **for** $i = 1, 2, \cdots, n$ **do**
6:         Query the dueling oracle with $(B_{\tau(i)}, B_{\tau(i)-1})$ and receive the feedback $o_{ti}$.
7:         Compute an unbiased estimator $\hat{g}_{ti} = \frac{o_{ti}}{a}$.
8:     **end for**
9:     Update $w^{(t+1)} = \Pi_{\mathcal{K}}(w^{(t)} - \eta \hat{g}_t)$.
10: **end for**
11: Set $\bar{w} = \frac{1}{T'} \sum_{t=1}^{T'} w^{(t)}$ and choose a threshold $z \in [0, 1]$ uniformly at random.
12: **return** $\hat{S}_T = \{i \mid \bar{w}_i \geq z\}$ .

---

## 4.2 Upper bound

We prove that Algorithm 1 achieves an error bound of $O\left(\frac{n^{\frac{3}{2}}}{a\sqrt{T}}\right)$ for any submodular objective function $f$. Similar proofs are given in Hazan & Kale (2012) and Ito (2019).

**Theorem 3.** *Let $f$ be a submodular function. Let $n, T$ and $a$ be the input of Algorithm 1. Then, Algorithm 1 with parameter $\eta = \frac{a}{2\sqrt{nT}}$ achieves the following error bound: $\mathbb{E}[E_T] = O\left(\frac{n^{\frac{3}{2}}}{a\sqrt{T}}\right)$. The expectation is taken with regard to the randomness of the oracle responses $o_t$, and the internal randomness of the algorithm.*

## 5 Sigmoid transfer function

In this section, we present an algorithm for submodular minimization using a dueling oracle when the transfer function is sigmoidal. This algorithm is similar to Algorithm 1 and performs SGD on the Lovász extension.

However, when the transfer function is nonlinear, it is not possible to construct an unbiased estimator of the subgradient of the Lovász extension from responses of the dueling oracle. To reduce the error introduced by SGD, it is necessary to design an estimator with small bias for the subgradient. We employ Firth's method to construct a small-bias estimator when the transfer function is sigmoidal, thereby mitigating the estimation error.

Firth (1993) has shown that in regular parametric problems, the first-order term of the asymptotic bias of the maximum likelihood estimates can be eliminated by penalizing the log likelihood. In particular, if $\theta$ is the canonical parameter of an exponential family model, the penalized log likelihood becomes $\frac{\partial}{\partial\theta} \log L(\theta) + \frac{1}{2}\frac{\partial}{\partial\theta} \log |I(\theta)|$, where $I(\theta)$ denotes the Fisher information evaluated at $\theta$.

**Proposition 3.** *Consider a logistic regression model:*

$$\Pr(X_i = +1 \mid \theta) = \frac{1}{1 + e^{-b\theta}}, \ \Pr(X_i = -1) = 1 - \Pr(X_i = +1 \mid \theta) = \frac{e^{-b\theta}}{1 + e^{-b\theta}}$$

*with $i = 1, \cdots, k$, $X_i \in \{\pm 1\}$ denoting the binary outcome variable and $b$ being a positive constant. The penalized maximum likelihood estimator $\hat{\theta}^*$ for the regression parameter $\theta \in [-1, 1]$ can be written as: $\hat{\theta}^* = \frac{1}{b} \log\left(\frac{k_+ + \frac{1}{2}}{k_- + \frac{1}{2}}\right)$, where $k_+ = |\{i \mid X_i = +1\}|$ and $k_- = |\{i \mid X_i = -1\}|$. Then, the bias of $\hat{\theta}^*$ satisfies: $|\mathbb{E}[\hat{\theta}^*] - \theta| \leq \left|\frac{2\psi - 1}{24b\psi^2(1-\psi)^2}\right| \frac{1}{k^2} + O\left(\frac{1}{k^3}\right)$. Here, $\psi$ is a constant and $\psi = \frac{1}{1+e^{-b}}$. We denote the coefficient of $\frac{1}{k^2}$ by $C(b) = \left|\frac{2\psi - 1}{24b\psi^2(1-\psi)^2}\right|$.*

---

**Algorithm 2** Submodular Stochastic Gradient Descent for sigmoid transfer function

---

**Input:** The size $n \geq 1$ of the ground set, the number $T \geq 1$ of oracle calls and the constant $0 < b$ of the transfer function $\rho(x) = 2/(1 + e^{-bx}) - 1$.

1: Set the initial point $w^{(1)} = 1/2 \cdot \mathbf{1}$,
   the constant $\psi = 1/(1 + e^{-b})$ and $C(b) = |(2\psi - 1)/(24b\psi^2(1 - \psi)^2)|$,
   the learning rate $\eta = b^{\frac{6}{5}} C(b)^{\frac{1}{5}} n^{\frac{2}{5}} / T^{\frac{2}{5}}$,
   the number of steps $T' = T^{\frac{4}{5}} / n^{\frac{4}{5}} b^{\frac{2}{5}} C(b)^{\frac{2}{5}}$,
   and the number of query repetitions $k = b^{\frac{2}{5}} C(b)^{\frac{2}{5}} T^{\frac{1}{5}} / n^{\frac{1}{5}}$.

2: **for** $t = 1, 2, \ldots, T'$ **do**
3:     Find a permutation $\pi$ corresponding to $w^{(t)}$, i.e., $w^{(t)}_{\pi(1)} \geq \cdots \geq w^{(t)}_{\pi(n)}$, and its inverse permutation $\tau$.
4:     Define $B_i = \{\pi(1), \ldots, \pi(i)\}$ for $i \in [n]$ and $B_0 = \varnothing$.
5:     **for** $i = 1, 2, \cdots, n$ **do**
6:         Repeatedly query the dueling oracle with $(B_{\tau(i)}, B_{\tau(i)-1})$ for $k$ times and receive feedback.
7:         Let $k_+$ and $k_-$ be the number of times $+1$ and $-1$ are returned.
8:         Compute an estimator of the subgradient $\hat{g}_{ti} = \frac{1}{b} \log\left(\frac{k_+ + \frac{1}{2}}{k_- + \frac{1}{2}}\right)$.
9:     **end for**
10:    Update $w^{(t+1)} = \Pi_{\mathcal{K}}(w^{(t)} - \eta \hat{g}_t)$.
11: **end for**
12: Set $\bar{w} = \frac{1}{T'} \sum_{t=1}^{T'} w^{(t)}$ and choose a threshold $z \in [0, 1]$ uniformly at random.
13: **return** $\hat{S}_T = \{i \mid \bar{w}_i \geq z\}$ .

---

## 5.1 ALGORITHM

When the transfer function $\rho$ is sigmoidal, querying $(B_{\tau(i)}, B_{\tau(i)-1})$ produces a dueling oracle response following a logistic regression model with a parameter $g_i$. From Proposition 3, estimating with Firth's method enables us to obtain low-bias estimators of subgradients. This algorithm can efficiently perform SGD even if unbiased estimators of subgradients cannot be constructed.

## 5.2 UPPER BOUND

In this subsection, we prove that Algorithm 2 achieves an error bound $E_T = O\left(\frac{C(b)^{\frac{1}{5}}}{b^{\frac{4}{5}}} \cdot \frac{n^{\frac{7}{5}}}{T^{\frac{2}{5}}}\right)$ for any submodular objective function $f$. The proof structure closely follows that of Theorem 3, with necessary adjustments to account for the sigmoid transfer function.

**Theorem 4.** *Let $f$ be a submodular function. Let $n, T$ and $b$ be the input of Algorithm 2. Then, Algorithm 2 achieves the following error bound:* $\mathbb{E}[E_T] = O\left(\frac{C(b)^{\frac{1}{5}}}{b^{\frac{4}{5}}} \cdot \frac{n^{\frac{7}{5}}}{T^{\frac{2}{5}}}\right)$ *. The expectation is taken with regard to the randomness of the oracle responses $o_t$, and the internal randomness of the algorithm.*

The factor $\frac{C(b)^{\frac{1}{5}}}{b^{\frac{4}{5}}}$ diverges to $+\infty$, as $b \to +0$ or $b \to +\infty$. Therefore, this algorithm cannot limit the error when $b$ takes extreme values.

## 6 LOWER BOUND

This section establishes lower bounds for the submodular minimization problem using a dueling oracle with linear or sigmoid transfer functions. Specifically, we analyze the following two scenarios.

1. A lower bound for algorithms that satisfy the following Restriction 1.
2. A general lower bound for any algorithm.

**Restriction 1.** *The symmetric difference between the two sets in each query remains constant order. Specifically, for any query $(S_t, S'_t)$, the condition $|S_t \triangle S'_t| := |(S_t \backslash S'_t) \cup (S'_t \backslash S_t)| = O(1)$ holds.*

Algorithms 1 and 2 satisfy this restriction. In fact, the pairs queried in the algorithms are restricted to those of the form $(B_{\tau(i)}, B_{\tau(i)-1})$, and by the definition of $B_i$ and $\tau$, we have $|B_{\tau(i)} \triangle B_{\tau(i)-1}| = |\{i\}| = 1$.

We prove that for algorithms that satisfy Restriction 1, there exists an instance of the problem where the error is at least $\Omega\left(\frac{n^{\frac{3}{2}}}{\sqrt{T}}\right)$. Additionally, for algorithms without any restrictions, we construct an instance where the error lower bound is $\Omega\left(\frac{n}{\sqrt{T}}\right)$. Since Algorithm 1 satisfies Restriction 1, it is optimal up to a constant factor among algorithms restricted by this condition.

**Theorem 5.** *In SFM using a dueling oracle with linear or sigmoid transfer functions, there exists an instance for which algorithms that satisfy Restriction 1 suffer an error of: $\mathbb{E}[E_T] = \Omega\left(\frac{n^{\frac{3}{2}}}{\sqrt{T}}\right)$. In addition, there is an instance for which algorithms without any restrictions suffer an error of: $\mathbb{E}[E_T] = \Omega\left(\frac{n}{\sqrt{T}}\right)$. The expectation is taken with regard to the randomness of the instance $f$ and oracles $o_t$, and the internal randomness of the algorithm.*

## 7    CONCLUSION AND OPEN QUESTIONS

We have presented algorithms with upper bounds and lower bounds for submodular function minimization using dueling oracles with linear or sigmoid transfer functions. In the case of linear transfer functions, the upper and lower bounds coincide in their dependence on $T$, establishing algorithmic optimality. By contrast, for nonlinear transfer functions, there remains a gap between the upper and lower bounds in terms of $T$-dependence, leaving room for improvement; we conjecture that the upper bound can be tightened to $O(\frac{1}{\sqrt{T}})$, which we leave as future work. Furthermore, we believe that a more detailed investigation into the dependence on $n$ in the linear case is also an important direction for future research. Relaxing the assumption that the transfer function is known is also an important direction for future work, and techniques in dueling convex optimization (Saha et al., 2025) and derivative-free optimization (Jamieson et al., 2012) might provide an effective approach to this issue. Furthermore, while the algorithm in this paper is designed via an extension to a continuous space, it is possible that a combinatorial approach, i.e., one that does not rely on continuous relaxations such as FTPL-based algorithms by Hazan & Kale (2012), may lead to a more efficient method. Exploring such alternative approaches constitutes another interesting direction for future research.

**Reproducibility statement**   The code necessary to reproduce the numerical experiments reported in Appendix B is included in the supplementary material. For the theoretical results, all claims without external references are accompanied by complete proofs provided either in the main text or in the appendix. These materials, together with the descriptions of assumptions and algorithmic details in the paper, are intended to ensure the reproducibility of our results.

## ACKNOWLEDGMENTS

We thank Takeru Matsuda for pointing us to the literature on Firth's method. SI was supported by JSPS KAKENHI Grant Number JP25K03184 and by JST PRESTO, Japan, Grant Number JPMJPR2511.

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

APPENDIX

## A  OMITTED PROOFS

**Proposition 1.** *Let $\hat{f}$ be a Lovász extension of a function $f : 2^{[n]} \to [0,1]$. For any $w \in \mathcal{K}$, the following equations hold:*

$$\hat{f}(w) = \int_0^1 f(\{i \mid w_i \geq z\})\mathrm{d}z$$
$$= \mathbb{E}_{z \sim \mathrm{Unif}([0,1])}[f(\{i \mid w_i \geq z\})].$$

*Proof.* Since the permutation $\pi$ is defined so that $(1 \geq)w_{\pi(1)} \geq \cdots \geq w_{\pi(n)}(\geq 0)$, the following equations follow:

$$\int_0^1 f(\{w \geq z\})\mathrm{d}z$$
$$= \int_0^{w_{\pi(n)}} f(\{w \geq z\})\mathrm{d}z + \cdots + \int_{w_{\pi(i+1)}}^{w_{\pi(i)}} f(\{w \geq z\})\mathrm{d}z + \cdots + \int_{w_{\pi(1)}}^1 f(\{w \geq z\})\mathrm{d}z$$
$$= f([n])(w_{\pi(n)} - 0) + \cdots + f(\{\pi(1), \ldots, \pi(i)\})(w_{\pi(i)} - w_{\pi(i+1)}) + \cdots + f(\varnothing)(1 - w_{\pi(1)})$$
$$= \sum_{i=1}^{n-1} f(\{\pi(1), \ldots, \pi(i)\})(w_{\pi(i)} - w_{\pi(i+1)}) + f([n])w_{\pi(n)} + f(\varnothing)(1 - w_{\pi(1)})$$
$$= \hat{f}(w).$$

If $z$ is chosen uniformly at random from $[0,1]$, the probability density function of a random variable $z$ is 1. Then, we have:

$$\mathbb{E}_{z \sim \mathrm{Unif}([0,1])}[f(\{w \geq z\})] = \int_0^1 f(\{w \geq z\})1\mathrm{d}z$$
$$= \hat{f}(w).$$

$\square$

**Theorem 1.** *(Fujishige (1991)) A function $f : 2^{[n]} \to [0,1]$ is submodular if and only if its Lovász extension $\hat{f} : \mathcal{K} \to [0,1]$ is convex. For a submodular function $f : 2^{[n]} \to [0,1]$ and its Lovász extension $\hat{f} : \mathcal{K} \to [0,1]$, we have:*

$$\min_{X \in 2^{[n]}} f(X) = \min_{w \in \{0,1\}^n} \hat{f}(w) = \min_{w \in [0,1]^n} \hat{f}(w).$$

To begin with, we prove the first part of Theorem 1, which states that a function $f$ is submodular if and only if its Lovász extension is convex. To prove this, we first define the submodular polyhedron in $\mathbb{R}^n$. We then consider a linear programming problem (LP) over the submodular polyhedron and apply the *strong duality* theorem to this LP. For $\chi_X \in \mathbb{R}^n$, we use the notation $(\chi_X)_i = \mathbb{1}\{i \in X\}$.

**Definition 2.** Let $f$ be a submodular function. The submodular polyhedron $P(f)$ is defined as:

$$P(f) = \{s \in \mathbb{R}^n \mid \forall X \in 2^{[n]}, s^\top \chi_X \leq f(X)\}.$$

**Theorem 6 (Strong Duality Theorem).** *Let $x, c \in \mathbb{R}^n$, $y, b \in \mathbb{R}^m$, $A \in \mathbb{R}^{m \times n}$. Consider the following primal problem and its corresponding dual problem in linear programming:*

- *Primal Problem:*

$$\begin{aligned} \text{maximize} \quad & b^\top y \\ \text{subject to} \quad & A^\top y \leq c \end{aligned}$$

- **_Dual Problem:_**

$$
\begin{aligned}
\text{minimize} \quad & c^\top x \\
\text{subject to} \quad & Ax = b \\
& x \geq 0
\end{aligned}
$$

*For these problems, if either the primal or dual problem has an optimal solution, then the optimal values of both problems are equal. That is,*

$$
\max\{b^\top y \mid A^\top y \geq c\} = \min\{c^\top x \mid Ax = b, x \geq 0\}.
$$

The following proposition, given in Bach (2011, Proposition 3.2), serves as a key building block for proof of the first part of Theorem 1:

**Proposition 4.** *Let $f$ be a submodular function. Let $w \in \mathcal{K}$, $\pi : [n] \to [n]$ be a permutation that orders the components of $w$ in decreasing order, and define $s_{\pi(i)} = f(\{\pi(1), \ldots, \pi(i)\}) - f(\{\pi(1), \ldots, \pi(i-1)\})$ for $i \in 2, \cdots, n$ and $s_{\pi(1)} = f(\{\pi(1)\}) - f(\varnothing)$. Then $s \in P(f)$ and,*

$$
s = \arg\max_{s \in P(f)} w^\top s,
$$

$$
\hat{f}(w) - f(\varnothing) = \max_{s \in P(f)} w^\top s.
$$

*Proof.* We consider the LP:

$$
\max_{s \in P(f)} w^\top s.
$$

By the definition of $P(f)$, the primal problem can be explicitly written as:

$$
\begin{aligned}
\text{maximize} \quad & w^\top s \\
\text{subject to} \quad & \chi_X^\top s \leq f(X) \quad \text{for all } X \in 2^{[n]}
\end{aligned}
$$

Let $\lambda_X$ be a real number for all $X \in 2^{[n]}$. The dual problem corresponding to this primal problem can be written as:

$$
\begin{aligned}
\text{minimize} \quad & \sum_{X \in 2^{[n]}} \lambda_X f(X) \\
\text{subject to} \quad & \sum_{X \in 2^{[n]}} \lambda_X \chi_X = w \\
& \lambda_X \geq 0 \quad \text{for all } X \in 2^{[n]}
\end{aligned}
$$

A constant vector, where each component is $\min_{X \in 2^{[n]}, X \neq \varnothing} \frac{f(X)}{|X|}$, belongs to $P(f)$, ensuring that $P(f)$ is non-empty and the primal problem has an optimal solution. Therefore, by the strong duality (Theorem 6), the optimal values of both problems are equal.

We define as:

$$
s_{\pi(i)} = \begin{cases} f(\{\pi(1)\}) - f(\varnothing) & (\text{if } i = 1), \\ f(\{\pi(1), \ldots, \pi(i)\}) - f(\{\pi(1), \ldots, \pi(i-1)\}) & (\text{if } i \in \{2, \cdots, n\}), \end{cases}
$$

$$
\lambda_X = \begin{cases} w_{\pi(i)} - w_{\pi(i+1)} & (\text{if } X = \{\pi(1), \cdots, \pi(i)\} \text{ for } i \in \{1, \cdots, n-1\}), \\ w_{[n]} & (\text{if } X = [n]), \\ 0 & (\text{otherwise}), \end{cases}
$$

and show that they are feasible solutions of primal and dual problems. Then, we show that they achieve the same objective value and are optimal solutions. Without loss of generality, we assume that $\pi(k) = k$ for all $k \in [n]$. For any set $X$, we have:

$$
\begin{aligned}
s^\top \chi_X &= \sum_{k=1}^{n} (\chi_X)_k s_k \\
&= \sum_{k=1}^{n} (\chi_X)_k [f(\{1, \ldots, k\}) - f(\{1, \ldots, k-1\})] \\
&\leq \sum_{k=1}^{n} (\chi_X)_k [f(X \cap \{1, \ldots, k\}) - f(X \cap \{1, \ldots, k-1\})] \quad \text{by the submodularity of } f, \\
&= \sum_{k=1}^{n} [f(X \cap \{1, \ldots, k\}) - f(X \cap \{1, \ldots, k-1\})] \\
&= f(X) \quad .
\end{aligned}
$$

Thus, $s \in P(f)$, i.e., $s$ is a feasible solution of the primal problem. $\lambda_X$ are all non-negative according to the definition of $\pi$, and satisfy the constraint $\sum_{X \in 2^{[n]}} \lambda_X \chi_X = w$. Therefore, $\lambda_X$ are feasible solutions of the dual problem. Since

$$
\begin{aligned}
w^\top s &= \sum_{i=1}^{n} w_{\pi(i)} s_{\pi(i)} \\
&= \sum_{i=2}^{n} w_{\pi(i)} [f(\{\pi(1), \cdots, \pi(i)\}) - f(\{\pi(1), \cdots, \pi(i-1)\})] \\
&\quad + w_{\pi(1)} [f(\{\pi(1)\}) - f(\varnothing)] \\
&= \hat{f}(w) - f(\varnothing),
\end{aligned}
$$

$$
\begin{aligned}
\sum_{X \in 2^{[n]}} \lambda_X f(X) &= \sum_{i=1}^{n-1} f(\{\pi(1), \ldots, \pi(i)\})(w_{\pi(i)} - w_{\pi(i+1)}) \\
&\quad + f([n]) w_{\pi(n)} - f(\varnothing) w_{\pi(1)} \\
&= \hat{f}(w) - f(\varnothing),
\end{aligned}
$$

both the primal and dual problems achieve the value $\hat{f}(w) - f(\varnothing)$. Thus, by strong duality, this is the optimal solution of both problems and $s, \lambda_X$ are the optimal solutions for the primal and dual problems, respectively. $\qquad \square$

Then, we prove the first part of Theorem 1 using this proposition.

*Proof.* If $f$ is a submodular function, from Proposition 4, for all $w \in \mathcal{K}$, $\hat{f}(w) - f(\varnothing)$ is a maximum of linear functions, and hence $\hat{f}(w) - f(\varnothing)$ is a convex function, i.e., $\hat{f}(w)$ is a convex function. Conversely, suppose that the Lovász extension $\hat{f}$ of $f$ is a convex function. By definition, $\hat{f}$ is a positively homogeneous function, i.e., $\hat{f}(\lambda w) = \lambda \hat{f}(w)$ for any $\lambda \in (0, 1]$. For any $X, Y \in 2^{[n]}$,

$$
\begin{aligned}
\hat{f}(\tfrac{1}{2}\chi_X + \tfrac{1}{2}\chi_Y) &= \hat{f}(\tfrac{1}{2}\chi_{X \cup Y} + \tfrac{1}{2}\chi_{X \cap Y}) \\
&= \hat{f}(\tfrac{1}{2}\chi_{X \cup Y}) + \hat{f}(\tfrac{1}{2}\chi_{X \cap Y}) \\
&= \tfrac{1}{2}\hat{f}(\chi_{X \cup Y}) + \tfrac{1}{2}\hat{f}(\chi_{X \cap Y}) \\
&= \tfrac{1}{2}f(X \cup Y) + \tfrac{1}{2}f(X \cap Y).
\end{aligned}
$$

The second equation holds because the components of each vector $\frac{1}{2}\chi_{X \cup Y} + \frac{1}{2}\chi_{X \cap Y}, \frac{1}{2}\chi_{X \cap Y}$ and $\chi_{X \cup Y}$ can be ordered in the same sequence and can share the same permutation $\pi$. Furthermore, since $\hat{f}$ is a convex function, we also have:

$$\hat{f}(\frac{1}{2}\chi_X + \frac{1}{2}\chi_Y) \leq \frac{1}{2}\hat{f}(\chi_X) + \frac{1}{2}\hat{f}(\chi_Y)$$
$$= \frac{1}{2}f(X) + \frac{1}{2}f(Y).$$

Therefore, $f$ is a submodular function. $\qquad\square$

Next, we establish the second part of Theorem 1, which states that the minimum value of a submodular function is equal to the minimum value of its Lovász extension.

*Proof.* Since $\hat{f}$ is an extension from $\{0,1\}^n$ to $[0,1]^n$ and satisfies $f(X) = \hat{f}(\chi_X)$ for all $X \in 2^{[n]}$, it follows that $\min_{X \in 2^{[n]}} f(X) = \min_{w \in \{0,1\}^n} \hat{f}(w) \geq \min_{w \in [0,1]^n} \hat{f}(w)$. Next, we prove the reverse inequality. From equation 2, the following holds for any $w \in [0,1]^n$:

$$\hat{f}(w) = \sum_{i=1}^{n-1} f(\{\pi(1), \ldots, \pi(i)\})(w_{\pi(i)} - w_{\pi(i+1)}) + f([n])w_{\pi(n)} + f(\varnothing)(1 - w_{\pi(1)})$$

$$\geq \sum_{i=1}^{n-1} \min_{X \in 2^{[n]}} f(X)(w_{\pi(i)} - w_{\pi(i+1)}) + \min_{X \in 2^{[n]}} f(X)w_{\pi(n)} + \min_{X \in 2^{[n]}} f(X)(1 - w_{\pi(1)})$$

$$= \min_{X \in 2^{[n]}} f(X)$$

$\qquad\square$

**Proposition 2.** *Let $f$ be a submodular function. For $w \in \mathcal{K}$, let $\pi$ be a permutation that orders the components of $w$ in decreasing order, and let $\tau$ be the inverse permutation of $\pi$. Then, a subgradient $g$ of $\hat{f}$ at $w$ is given as follows:*

$$g_{\pi(i)} = f(B_i) - f(B_{i-1})$$
$$g_i = f(B_{\tau(i)}) - f(B_{\tau(i)-1})$$

*Proof.* From equation 2, the expression for a subgradient is immediately obtained. $\qquad\square$

**Theorem 2.** *Let $\hat{f} : \mathcal{K} \to [0,1]$ be a convex function in the hypercube $\mathcal{K} = [0,1]^n$. Let $w^{(1)}, w^{(2)}, \ldots, w^{(T)}$ be defined by $w^{(1)} = \frac{1}{2} \cdot \mathbf{1}$ and $w^{(t+1)} = \Pi_{\mathcal{K}}(w^{(t)} - \eta\hat{g}_t)$. When $\hat{g}_1, \hat{g}_2, \ldots, \hat{g}_T$ are unbiased estimators of subgradients such that $\mathbb{E}[\hat{g}_t \mid w^{(t)}] = g_t$, where $g_t$ is a subgradient of $\hat{f}$ at $w^{(t)}$, the average point $\bar{w} = \frac{1}{T}\sum_{t=1}^{T} w^{(t)}$ satisfies:*

$$\mathbb{E}[\hat{f}(\bar{w})] - \min_{w^* \in \mathcal{K}} \hat{f}(w^*) \leq \frac{1}{T}\left(\frac{n}{8\eta} + \frac{\eta}{2}\sum_{t=1}^{T} \mathbb{E}[\|\hat{g}_t\|_2^2]\right).$$

*Proof.* Let $v^{(t+1)} = w^{(t)} - \eta\hat{g}_t$, so that $w^{(t+1)} = \Pi_{\mathcal{K}}(v^{(t+1)})$. By expanding the squared norm, we have:

$$\|v^{(t+1)} - w^*\|_2^2 = \|w^{(t)} - w^*\|_2^2 - 2\eta\hat{g}_t^\top(w^{(t)} - w^*) + \eta^2\|\hat{g}_t\|_2^2.$$

Rearranging terms, we obtain:

$$\hat{g}_t^\top(w^{(t)} - w^*) = \frac{1}{2\eta}[\|w^{(t)} - w^*\|_2^2 - \|v^{(t+1)} - w^*\|_2^2] + \frac{\eta}{2}\|\hat{g}_t\|_2^2.$$

Using the non-expansiveness property of Euclidean projections onto convex sets,

$$\|w^{(t+1)} - w^*\|_2^2 \leq \|v^{(t+1)} - w^*\|_2^2,$$

we further obtain:

$$\hat{g}_t^\top (w^{(t)} - w^*) \le \frac{1}{2\eta}[\|w^{(t)} - w^*\|_2^2 - \|w^{(t+1)} - w^*\|_2^2] + \frac{\eta}{2}\|\hat{g}_t\|_2^2,$$

Summing this inequality over $t = 1, \dots, T$, we have:

$$\sum_{t=1}^T \hat{g}_t^\top (w^{(t)} - w^*) \le \sum_{t=1}^T \frac{\|w^{(t)} - w^*\|_2^2 - \|w^{(t+1)} - w^*\|_2^2}{2\eta} + \frac{\eta}{2}\|\hat{g}_t\|_2^2$$

$$\le \frac{n}{8\eta} + \frac{\eta}{2}\sum_{t=1}^T \|\hat{g}_t\|_2^2, \tag{4}$$

since $\|w^{(1)} - w^*\|_2^2 \le \frac{n}{4}$ ($|w_i^{(1)} - w_i^*| \le \frac{1}{2}$ for all $i \in [n]$ since $w^{(1)} = \frac{1}{2} \cdot \mathbf{1}$ and $w^* \in [0,1]^n$). Next, since $\mathbb{E}[\hat{g}_t \mid w^{(t)}] = g_t$, a subgradient of $\hat{f}$ at $w^{(t)}$, we have:

$$\mathbb{E}[\hat{g}_t^\top (w^{(t)} - w^*) \mid w^{(t)}] = g_t^\top (w^{(t)} - w^*) \ge \hat{f}(w^{(t)}) - \hat{f}(w^*),$$

where the inequality follows from the convexity of $\hat{f}$. Taking the expectation over the choice of $w^{(t)}$, we have:

$$\mathbb{E}[\hat{g}_t^\top (w^{(t)} - w^*)] \ge \mathbb{E}[\hat{f}(w^{(t)})] - \hat{f}(w^*).$$

Summing over $t = 1, \dots, T$, we have:

$$\sum_{t=1}^T \mathbb{E}[\hat{f}(w^{(t)})] - \sum_{t=1}^T \hat{f}(w^*) \le \mathbb{E}[\sum_{t=1}^T \hat{g}_t^\top (w^{(t)} - w^*)]$$

$$\le \frac{n}{8\eta} + \frac{\eta}{2}\sum_{t=1}^T \mathbb{E}[\|\hat{g}_t\|_2^2].$$

Finally, since $\hat{f}$ is convex, we can apply Jensen's inequality to obtain the expected error bound as follows:

$$\mathbb{E}[\hat{f}(\bar{w})] - \hat{f}(w^*) = \mathbb{E}[\hat{f}(\frac{1}{T}\sum_{t=1}^T w^{(t)})] - \hat{f}(w^*)$$

$$\le \frac{1}{T}\sum_{t=1}^T \mathbb{E}[\hat{f}(w^{(t)})] - \hat{f}(w^*) \quad \text{(By Jensen's inequality)}$$

$$= \frac{1}{T}\Big(\sum_{t=1}^T \mathbb{E}[\hat{f}(w^{(t)})] - \sum_{t=1}^T \hat{f}(w^*)\Big)$$

$$\le \frac{1}{T}\Big(\frac{n}{8\eta} + \frac{\eta}{2}\sum_{t=1}^T \mathbb{E}[\|\hat{g}_t\|_2^2]\Big).$$

$\square$

**Theorem 3.** *Let $f$ be a submodular function. Let $n, T$ and $a$ be the input of Algorithm 1. Then, Algorithm 1 with parameter $\eta = \frac{a}{2\sqrt{nT}}$ achieves the following error bound:*

$$\mathbb{E}[E_T] = O\left(\frac{n^{\frac{3}{2}}}{a\sqrt{T}}\right).$$

*The expectation is taken with regard to the randomness of the oracle responses $o_t$, and the internal randomness of the algorithm.*

*Proof.* Since $\hat{g}_t$ in Algorithm 1 is an unbiased estimator of the subgradient $g_t$, by Theorem 2, the following inequality holds:

$$\mathbb{E}[\hat{f}(\bar{w})] - \min_{w^* \in \mathcal{K}} \hat{f}(w^*) \le \frac{1}{T}\Big(\frac{n}{8\eta} + \frac{\eta}{2}\sum_{t=1}^T \mathbb{E}[\|\hat{g}_t\|_2^2]\Big).$$

Moreover, since $\hat{g}_t = \frac{n}{a} o_t \cdot e_i$ and $\|\hat{g}_t\|_2^2 = \frac{n^2}{a^2}$, we have:

$$\sum_{t=1}^T \mathbb{E}[\|\hat{g}_t\|_2^2] = \frac{n^2 T}{a^2}.$$

Then, we obtain:

$$\mathbb{E}[\hat{f}(\bar{w})] - \min_{w^* \in \mathcal{K}} \hat{f}(w^*) \leq \frac{1}{T}\left(\frac{n}{8\eta} + \frac{\eta}{2}\sum_{t=1}^T \mathbb{E}[\|\hat{g}_t\|_2^2]\right)$$

$$= \frac{n}{8\eta T} + \frac{\eta n^2}{2a^2}$$

$$= \frac{n^{\frac{3}{2}}}{4a\sqrt{T}},$$

where the last equality follows by setting $\eta = \frac{a}{2\sqrt{nT}}$. Finally, from equation 3 and Theorem 1, we have the error bound:

$$\mathbb{E}[E_T] = \mathbb{E}[f(\hat{S}_T)] - \min_{S \in 2^{[n]}} f(S)$$

$$= \mathbb{E}[\hat{f}(\bar{w})] - \min_{w^* \in \mathcal{K}} \hat{f}(w^*)$$

$$\leq \frac{n^{\frac{3}{2}}}{4a\sqrt{T}}.$$

$\square$

**Proposition 3.** *Consider a logistic regression model:*

$$\Pr(X_i = +1 \mid \theta) = \frac{1}{1 + e^{-b\theta}}, \; \Pr(X_i = -1) = 1 - \Pr(X_i = +1 \mid \theta) = \frac{e^{-b\theta}}{1 + e^{-b\theta}}$$

*with $i = 1, \cdots, k$, $X_i \in \{\pm 1\}$ denoting the binary outcome variable and $b$ being a positive constant. The penalized maximum likelihood estimator $\hat{\theta}^*$ for the regression parameter $\theta \in [-1, 1]$ can be written as:*

$$\hat{\theta}^* = \frac{1}{b} \log\left(\frac{k_+ + \frac{1}{2}}{k_- + \frac{1}{2}}\right),$$

*where $k_+ = |\{i \mid X_i = +1\}|$ and $k_- = |\{i \mid X_i = -1\}|$. Then, the bias of $\hat{\theta}^*$ satisfies:*

$$|\mathbb{E}[\hat{\theta}^*] - \theta| \leq \left|\frac{2\psi - 1}{24b\psi^2(1 - \psi)^2}\right| \frac{1}{k^2} + O\left(\frac{1}{k^3}\right).$$

*Here, $\psi$ is a constant and $\psi = \frac{1}{1 + e^{-b}}$. We denote the coefficient of $\frac{1}{k^2}$ by $C(b) = \left|\frac{2\psi - 1}{24b\psi^2(1-\psi)^2}\right|$.*

First, we derive the form of the penalized maximum likelihood estimator, which constitutes the first part of this proposition.

*Proof.* The log likelihood function and its partial derivative with respect to $\theta$ are

$$\log L(\theta) = \sum_{i=1}^k \log\left(\frac{1}{1 + e^{-X_i b\theta}}\right)$$

$$= k_{+1} \log\left(\frac{1}{1 + e^{-b\theta}}\right) + k_{-1} \log\left(\frac{1}{1 + e^{b\theta}}\right),$$

$$\frac{\partial}{\partial\theta} \log L(\theta) = k_{+1}(1 + e^{-b\theta})\frac{be^{-b\theta}}{(1 + e^{-b\theta})^2} + k_{-1}(1 + e^{b\theta})\frac{-be^{b\theta}}{(1 + e^{b\theta})^2}$$

$$= k_{+1}b\frac{1}{1 + e^{b\theta}} - k_{-1}b\frac{1}{1 + e^{-b\theta}}.$$

The Fisher information is

$$I(\theta) = \mathbb{E}[-\frac{\partial^2}{\partial\theta^2} \log L(\theta) \mid \theta]$$

$$= \mathbb{E}[k_+ b^2 \frac{e^{b\theta}}{(1 + e^{b\theta})^2} + k_- b^2 \frac{e^{-b\theta}}{(1 + e^{-b\theta})^2} \mid \theta]$$

$$= kb^2 \frac{e^{-b\theta}}{(1 + e^{-b\theta})^2},$$

since $\mathbb{E}[k_+ \mid \theta] = \frac{k}{1+e^{-b\theta}}$ and $\mathbb{E}[k_- \mid \theta] = \frac{k}{1+e^{b\theta}}$. Therefore, the modified score function is

$$U^*(\theta) = \frac{\partial}{\partial\theta} \log L(\theta) + \frac{1}{2} \frac{\partial}{\partial\theta} \log |I(\theta)|$$

$$= k_+ b \frac{1}{1 + e^{b\theta}} - k_- b \frac{1}{1 + e^{-b\theta}} - \frac{b}{2} + \frac{be^{-b\theta}}{1 + e^{-b\theta}}$$

$$= \frac{b}{1 + e^{-b\theta}} \Big( (k_+ + \frac{1}{2}) e^{-b\theta} - (k_- + \frac{1}{2}) \Big).$$

Finally, we obtain the solution $\hat{\theta}^*$:

$$e^{-b\hat{\theta}^*} = \frac{k_- + \frac{1}{2}}{k_+ + \frac{1}{2}}$$

$$-b\hat{\theta}^* = \log \Big( \frac{k_- + \frac{1}{2}}{k_+ + \frac{1}{2}} \Big)$$

$$\hat{\theta}^* = \frac{1}{b} \log \Big( \frac{k_+ + \frac{1}{2}}{k_- + \frac{1}{2}} \Big).$$

$\square$

Next, we establish the second part of Proposition 3, which demonstrates the bias of $\hat{\theta}^*$. Our proof is based on the method outlined in Cox & Snell (1989, §2.1.6). Although their work showed that the leading bias term, proportional to $\frac{1}{k}$, can be eliminated, we extend this approach to derive terms up to $\frac{1}{k^2}$.

*Proof.* Let $\phi = \frac{1}{1+e^{-b\theta}}$, be the probability of $X_i = +1$. The parameter $\theta$ can be expressed as $\frac{1}{b} \log \frac{\phi}{1-\phi}$. Define a function $h : [0, 1] \to \mathbb{R}$ as $h(x) = \frac{1}{b} \log \frac{x + \frac{1}{2k}}{1 - x + \frac{1}{2k}}$. Then, $\hat{\theta}^* = h(\frac{k_+}{k})$. Considering the Taylor expansion of $h(\frac{k_+}{k})$ with respect to $\phi$, we have:

$$\hat{\theta}^* = h(\frac{k_+}{k}) = h(\phi) + h'(\phi)(\frac{k_+}{k} - \phi) + \frac{1}{2} h''(\phi)(\frac{k_+}{k} - \phi)^2$$

$$+ \frac{1}{6} h'''(\phi)(\frac{k_+}{k} - \phi)^3 + \frac{1}{24} h''''(\phi)(\frac{k_+}{k} - \phi)^4 + \cdots. \tag{5}$$

By the definition of $k_+$, $\mathbb{E}[\frac{k_+}{k}] = \phi$. Therefore, $\mathbb{E}[(\frac{k_+}{k} - \phi)^r]$ is the $r$-th central moment of a binomial distribution. Taking expectation of equation 5, we obtain:

$$
\begin{aligned}
b\mathbb{E}[\hat{\theta}^*] &= b\Big\{ h(\phi) + h'(\phi)\mathbb{E}[\frac{k_+}{k} - \phi] + \frac{1}{2}h''(\phi)\mathbb{E}[(\frac{k_+}{k} - \phi)^2] + \frac{1}{6}h'''(\phi)\mathbb{E}[(\frac{k_+}{k} - \phi)^3] \\
&\quad + \frac{1}{24}h''''(\phi)\mathbb{E}[(\frac{k_+}{k} - \phi)^4] + \cdots \Big\} \\
&= \log\frac{\phi + \frac{1}{2k}}{1 - \phi + \frac{1}{2k}} + \frac{1}{2}\left\{ -\left(\phi + \frac{1}{2k}\right)^{-2} + \left(1 - \phi + \frac{1}{2k}\right)^{-2} \right\}\frac{\phi(1-\phi)}{k} \\
&\quad + \frac{1}{6}\left\{ 2\left(\phi + \frac{1}{2k}\right)^{-3} + 2\left(1 - \phi + \frac{1}{2k}\right)^{-3} \right\}\frac{\phi(1-\phi)(1-2\phi)}{k^2} \\
&\quad + \frac{1}{24}\left\{ -6\left(\phi + \frac{1}{2k}\right)^{-4} + 6\left(1 - \phi + \frac{1}{2k}\right)^{-4} \right\}\left\{ \frac{3\phi^2(1-\phi)^2}{k^2} \right.\\
&\qquad\qquad\qquad\qquad\qquad\qquad\qquad \left. + \frac{\phi(1-\phi)(1-6\phi(1-\phi))}{k^3} \right\} + \cdots \\
&= \log\frac{\phi}{1-\phi} + \frac{1}{2\phi k} - \frac{1}{8\phi^2 k^2} - \frac{1}{2(1-\phi)k} + \frac{1}{8(1-\phi)^2 k^2} \\
&\quad + \frac{1}{2}\left\{ -\frac{1}{\phi^2}\left(1 - \frac{1}{\phi k}\right) + \frac{1}{(1-\phi)^2}\left(1 - \frac{1}{(1-\phi)k}\right) \right\}\phi(1-\phi)\frac{1}{k} \\
&\quad + \frac{1}{3}\left\{ \frac{1}{\phi^3} + \frac{1}{(1-\phi)^3} \right\}\phi(1-\phi)(1-2\phi)\frac{1}{k^2} \\
&\quad + \frac{3}{4}\left\{ -\frac{1}{\phi^4} + \frac{1}{(1-\phi)^4} \right\}\phi^2(1-\phi)^2\frac{1}{k^2} + O\left(\frac{1}{k^3}\right) \\
&= b\theta + \left\{ \frac{1}{2\phi} - \frac{1}{2(1-\phi)} - \frac{1-\phi}{2\phi} + \frac{\phi}{2(1-\phi)} \right\}\frac{1}{k} + \left\{ -\frac{1}{8\phi^2} + \frac{1}{8(1-\phi)^2} + \frac{1-\phi}{2\phi^2} \right.\\
&\quad \left. - \frac{\phi}{2(1-\phi)^2} + \frac{(1-\phi)(1-2\phi)}{3\phi^2} + \frac{\phi(1-2\phi)}{3(1-\phi)^2} - \frac{3(1-\phi)^2}{4\phi^2} + \frac{3\phi^2}{4(1-\phi)^2} \right\} + O\left(\frac{1}{k^3}\right) \\
&\qquad\qquad\qquad\qquad \text{(By the Taylor expansion of } \log(1+x) \text{ and } (1+x)^{-r}) \\
&= b\theta + \frac{2\phi - 1}{24\phi^2(1-\phi)^2} \cdot \frac{1}{k^2} + O\left(\frac{1}{k^3}\right).
\end{aligned}
$$

Thus, the bias of $\hat{\theta}^*$ can be written as:

$$
|\mathbb{E}[\hat{\theta}^*] - \theta| = \left| \frac{2\phi - 1}{24b\phi^2(1-\phi)^2} \right| \frac{1}{k^2} + O\left(\frac{1}{k^3}\right).
$$

Considering $\phi$ as a function of $\theta$ and $b$ as a function of $\phi(\theta)$ and $\theta$, the relationship between $\phi$ and $\theta$, and the relationship between the coefficient of the term $\frac{1}{k^2}$ and $\phi$ are illustrated in Figure 1 and Figure 2. For all $b > 0$, the coefficient reaches its maximum with $\theta = 1$ (or $\theta = -1$). Therefore, the coefficient can be bounded using $\psi = \frac{1}{1+e^{-b}}$:

$$
\left| \frac{2\phi - 1}{24b\phi^2(1-\phi)^2} \right| \leq \left| \frac{2\psi - 1}{24b\psi^2(1-\psi)^2} \right| = C(b).
$$

$\square$

**Theorem 4.** *Let $f$ be a submodular function. Let $n, T$ and $b$ be the input of Algorithm 2. Then, Algorithm 2 achieves the following error bound:*

$$
\mathbb{E}[E_T] = O\left( \frac{C(b)^{\frac{1}{5}}}{b^{\frac{4}{5}}} \cdot \frac{n^{\frac{7}{5}}}{T^{\frac{2}{5}}} \right).
$$

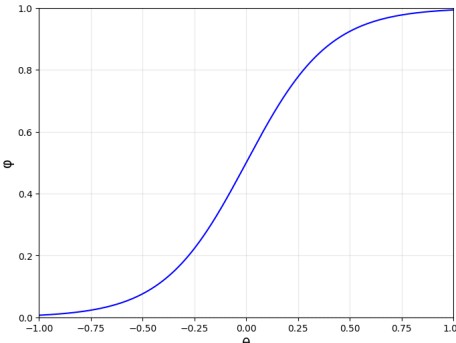

Figure 1: Relationship between $\phi$ and $\theta$.

Figure 2: Relationship between $|(2\phi - 1)/(24b\phi^2(1-\phi)^2)|$ and $\phi$.

*The expectation is taken with regard to the randomness of the oracle responses $o_t$, and the internal randomness of the algorithm.*

*Proof.* Let $w^*$ be a minimizer of $\hat{f}$ in $\mathcal{K}$.

$$\sum_{t=1}^{T'} \left( \hat{f}(w^{(t)}) - \hat{f}(w^*) \right) \leq \sum_{t=1}^{T'} \left( g_t^\top (w^{(t)} - w^*) \right)$$

$$= \sum_{t=1}^{T'} \left( \hat{g}_t^\top (w^{(t)} - w^*) + (g_t^\top - \hat{g}_t^\top)(w^{(t)} - w^*) \right)$$

$$\leq \frac{n}{8\eta} + \frac{\eta}{2} \sum_{t=1}^{T'} \|\hat{g}_t\|_2^2 + \sum_{t=1}^{T'} (g_t^\top - \hat{g}_t^\top)(w^{(t)} - w^*) \quad \text{(By equation 4)}$$

$$\leq \frac{n}{8\eta} + \frac{\eta}{2} \sum_{t=1}^{T'} \|\hat{g}_t\|_2^2 + \sum_{t=1}^{T'} \|g_t^\top - \hat{g}_t^\top\|_1 \|w^{(t)} - w^*\|_\infty$$

$$\text{(By Hölder's inequality)}$$

$$\leq \frac{n}{8\eta} + \frac{\eta}{2} \sum_{t=1}^{T'} \|\hat{g}_t\|_2^2 + \sum_{t=1}^{T'} \|g_t^\top - \hat{g}_t^\top\|_1.$$

The last inequality holds since $w^{(t)}, w^* \in \mathcal{K}$. By the definition of $\hat{g}_t$, we have:

$$\|\hat{g}_t\|_2^2 \leq \frac{n}{b^2} \{\log(2k+1)\}^2, \ \|\hat{g}_t - g_t\|_1 = nC(b)O\left(\frac{1}{k^2}\right).$$

Then, the following inequality follows by taking expectations and using Jensen's inequality:

$$\mathbb{E}[\hat{f}(\bar{w})] - \min_{w^* \in \mathcal{K}} \hat{f}(w^*) \leq \frac{1}{T'} \left( \frac{n}{8\eta} + \frac{\eta}{2} \sum_{t=1}^{T'} \|\hat{g}_t\|_2^2 + \sum_{t=1}^{T'} \|g_t^\top - \hat{g}_t^\top\|_1 \right)$$

$$\leq \frac{n}{8T'\eta} + \frac{n\eta}{2b^2} \{\log(2k+1)\}^2 + nC(b)O\left(\frac{1}{k^2}\right)$$

$$= \frac{n^2 k}{8T\eta} + \frac{n\eta}{2b^2} \{\log(2k+1)\}^2 + nC(b)O\left(\frac{1}{k^2}\right).$$

Setting $\eta = b^{\frac{6}{5}} C(b)^{\frac{1}{5}} \frac{n^{\frac{2}{5}}}{T^{\frac{2}{5}}}$ and $k = b^{\frac{2}{5}} C(b)^{\frac{2}{5}} \frac{T^{\frac{1}{5}}}{n^{\frac{1}{5}}}$, we have:

$$\mathbb{E}[\hat{f}(\bar{w})] - \min_{w^* \in \mathcal{K}} \hat{f}(w^*) = O\left( \frac{C(b)^{\frac{1}{5}}}{b^{\frac{4}{5}}} \cdot \frac{n^{\frac{7}{5}}}{T^{\frac{2}{5}}} \right).$$

Finally, from equation 3 and Theorem 1, we have the error bound:

$$\mathbb{E}[E_T] = \mathbb{E}[f(\hat{S}_T)] - \min_{S \in 2^{[n]}} f(S)$$

$$= \mathbb{E}[\hat{f}(\bar{w})] - \min_{w^* \in \mathcal{K}} \hat{f}(w^*)$$

$$= O\left( \frac{C(b)^{\frac{1}{5}}}{b^{\frac{4}{5}}} \cdot \frac{n^{\frac{7}{5}}}{T^{\frac{2}{5}}} \right).$$

$\square$

**Theorem 5.** *In SFM using a dueling oracle with linear or sigmoid transfer functions, there exists an instance for which algorithms that satisfy Restriction 1 suffer an error of:*

$$\mathbb{E}[E_T] = \mathbb{E}[f(\hat{S}_T) - \min_{S \in 2^{[n]}} f(S)] = \Omega\left( \frac{n^{\frac{3}{2}}}{\sqrt{T}} \right).$$

*In addition, there is an instance for which algorithms without any restrictions suffer an error of:*

$$\mathbb{E}[E_T] = \mathbb{E}[f(\hat{S}_T) - \min_{S \in 2^{[n]}} f(S)] = \Omega\left( \frac{n}{\sqrt{T}} \right).$$

*The expectation is taken with regard to the randomness of the instance $f$ and oracles $o_t$, and the internal randomness of the algorithm.*

Before presenting the proof of the lower bound, we introduce Yao's principle, a powerful tool for analyzing lower bounds of randomized algorithms. This principle allows us to derive lower bounds by analyzing the expected performance of deterministic algorithms over a chosen distribution of problem instances.

To begin with, we prove the lower bound for the linear transfer function.

**Proposition 4** (**Yao's principle**). *The worst-case error of any randomized algorithm is at least the error of the best deterministic algorithm under a specific distribution $\mathcal{D}$. That is, let the error $E_T$ be expressed as a function $E_T(A, x)$, where $A$ is an algorithm (deterministic or randomized) and $x$ represents the input (including the objective function $f$, oracle response $o_t$). Define $\mathcal{A}$ as the set of all deterministic algorithms, $\mathcal{R}$ as the set of all randomized algorithms, $\mathcal{X}$ as the set of all possible inputs, and $\mathcal{D}$ as a specific input distribution. Then, the following inequality holds:*

$$\min_{A \in \mathcal{A}} \mathbb{E}_{x \sim \mathcal{D}}[E_T(A, x)] \leq \max_{x \in \mathcal{X}} \mathbb{E}[E_T(R, x)].$$

*Proof.* For simplicity, this proof considers the objective function to be a submodular function $f : 2^{[n]} \to [-1, 1]$. Since submodular functions remain submodular under scaling by a constant or adding/subtracting a constant, this proof is applicable to the problem setting as well.

First, we consider the case where the algorithm satisfies Restriction 1. By Proposition 4, to establish a lower bound, we construct an objective function in which any deterministic algorithm (satisfies Restriction 1) incurs an error of $\Omega\left( \frac{n^{\frac{3}{2}}}{a\sqrt{T}} \right)$.

Fix a subset $S^* \in 2^{[n]}$ and a positive real value $\epsilon \in [0, 1]$. Define a submodular function $f : 2^{[n]} \to [-1, 1]$ as $f(X) = \frac{\epsilon}{2n}(|X \backslash S^*| - |X \cap S^*|)$. When a new element $i$ is added to a subset, the increment in the function value only depends on whether $i$ is included in $S^*$. Hence, as equation 1 holds, $f$ defined as above is a submodular function. It is obvious that $f$ achieves its minimum value of $-\frac{\epsilon}{2n}|S^*|$ at $S^*$. Thus, the error of the output $\hat{S}_T$ is given by the following expression:

$$f(\hat{S}_T) - f(S^*) = \frac{\epsilon}{2n}(|\hat{S}_T \backslash S^*| - |\hat{S}_T \cap S^*|) - (-\frac{\epsilon}{2n}|S^*|)$$

$$= \frac{\epsilon}{2n}|\hat{S}_T \triangle S^*|.$$

According to the restriction, the query at iteration $t$ is given by $(S_t + i, S_t)$ $(i \notin S_t)$. Based on the definition of $f$, the result $o_t$ of the dueling oracle follows the probability distribution:

$$
o_t \sim \begin{cases} \text{Ber}^\pm(-a\frac{\epsilon}{2n}) & \text{if } i \in S^* \\ \text{Ber}^\pm(a\frac{\epsilon}{2n}) & \text{if } i \notin S^*. \end{cases}
$$

Since the output $o_t$ does not depend on $S_t$ but only depends on $i_t$, let $i_t$ represent the query in iteration $t$.

Since we consider deterministic algorithms, the query in iteration $t$ is determined by $(i_s, o_s)_{s=1}^{t-1}$, and the output $\hat{S}_T$ is determined by $(i_t, o_t)_{t=1}^T$. Therefore, if $S^*$ is fixed, the output $\hat{S}_T$ can be considered as being sampled from a probability distribution that depends on $S^*$.

To simplify the proof, we define $\hat{x} = \chi_{\hat{S}_T}$ and $x^* = \chi_{S^*}$. Then, the error can be expressed using $\hat{x}$ and $x^*$ as $\frac{\epsilon}{2n}|\hat{S}_T \triangle S^*| = \frac{\epsilon}{2n} \sum_{i=1}^n |\hat{x}_i - x_i^*|$. Given that $\hat{x}$ follows a probability distribution $D_{x^*}$ determined by $x^*$, we consider the error as a function of $x^*$. Specifically, we define:

$$
E_T(x^*) = \mathbb{E}_{\hat{x} \sim D_{x^*}} [\frac{\epsilon}{2n} \sum_{i=1}^n |\hat{x}_i - x_i^*|].
$$

First, we bound the KL divergence between the probability distributions followed by $\hat{S}_T$ when $S^*$ and $S^* \triangle \{i\}$ are respectively given. By the definition of $\hat{x}$ and $x^*$, we consider the KL divergence between the distribution $D_{x^*}$, which $\hat{x}$ follows when $x^*$ is given, and the distribution $D_{x^* \triangle \{i\}}$, which $\hat{x}$ follows when $x^* \triangle \{i\}$ is given. By the *chain rule* for KL divergence [Cover & Thomas (2005, Theorem 2.5.3)], we have:

$$
\begin{aligned}
D_{\text{KL}}(D_{x^*} \| D_{x^* \triangle \{i\}}) &= D_{\text{KL}}(\hat{x}|_{x^*} \| \hat{x}_{x^* \triangle \{i\}}) \\
&\leq D_{\text{KL}}((i_t, o_t)_{t=1}^T|_{x^*} \| (i_t, o_t)_{t=1}^T|_{x^* \triangle \{i\}}) \\
&= D_{\text{KL}}((i_t, o_t)_{t=1}^{T-1}|_{x^*} \| (i_t, o_t)_{t=1}^{T-1}|_{x^* \triangle \{i\}}) \\
&\quad + \mathbb{E}_{(i_t, o_t)_{t=1}^{T-1}|_{x^*}} [D_{\text{KL}}((i_T, o_T)|_{x^*} \| (i_T, o_T)|_{x^* \triangle \{i\}})] \\
&\qquad\qquad\qquad\qquad\qquad\qquad \text{(By the chain rule)} \\
&= D_{\text{KL}}((i_t, o_t)_{t=1}^{T-1}|_{x^*} \| (i_t, o_t)_{t=1}^{T-1}|_{x^* \triangle \{i\}}) \\
&\quad + \mathbb{E}_{(i_t, o_t)_{t=1}^{T-1}|_{x^*}} [\mathbb{1}\{i_T = i\} D_{\text{KL}}(\text{Ber}^\pm(a\frac{\epsilon}{2\text{n}}) \| \text{Ber}^\pm(-a\frac{\epsilon}{2\text{n}}))] \\
&= D_{\text{KL}}((i_t, o_t)_{t=1}^{T-1}|_{x^*} \| (i_t, o_t)_{t=1}^{T-1}|_{x^* \triangle \{i\}}) \\
&\quad + \text{Pr}_{x^*}(i_T = i)(a\frac{\epsilon}{2n} \log(\frac{1 + a\frac{\epsilon}{2n}}{1 - a\frac{\epsilon}{2n}})) \\
&\qquad\qquad\qquad\qquad\qquad \text{(By the definition of KL divergence)} \\
&\leq \sum_{t=1}^T \text{Pr}_{x^*}(i_t = i)(a\frac{\epsilon}{2n} \log(\frac{1 + a\frac{\epsilon}{2n}}{1 - a\frac{\epsilon}{2n}})).
\end{aligned}
$$

The last inequality holds by applying the chain rule repeatedly. Since $\sum_{i=1}^n \sum_{t=1}^T \text{Pr}_{x^*}(i_t = i) = T$, we have the following inequality:

$$
\begin{aligned}
\sum_{i=1}^n D_{\text{KL}}(D_{x^*} \| D_{x^* \triangle \{i\}}) &\leq Ta\frac{\epsilon}{2n} \log(\frac{1 + a\frac{\epsilon}{2n}}{1 - a\frac{\epsilon}{2n}}) \\
&\leq \frac{\log 3}{2} \cdot \frac{a^2 T}{n^2} \epsilon^2.
\end{aligned} \tag{6}
$$

The last inequality holds, since $\frac{x}{2} \log(\frac{1+\frac{x}{2}}{1-\frac{x}{2}}) \leq \frac{\log 3}{2} x^2$ for $0 < x \leq 1$, and $0 < a\frac{\epsilon}{n} \leq 1$.

Next, we define $y_i(x^*) = \mathbb{E}_{\hat{x} \sim D_{x^*}}[\hat{x}_i]$ and $e_i(x^*) = |y_i(x^*) - x_i^*|$. since $x_i^*$ only takes values in $\{0, 1\}$, the error $E_T(x^*)$ can be expressed as:

$$
\begin{aligned}
E_T(x^*) &= \frac{\epsilon}{2n} \sum_{i=1}^{n} \mathbb{E}_{\hat{x} \sim D_{x^*}} |\hat{x}_i - x_i^*| \\
&= \frac{\epsilon}{2n} \sum_{i=1}^{n} |\mathbb{E}_{\hat{x} \sim D_{x^*}}[\hat{x}_i] - x_i^*| \\
&= \frac{\epsilon}{2n} \sum_{i=1}^{n} |y_i(x^*) - x_i^*| \\
&= \frac{\epsilon}{2n} \sum_{i=1}^{n} e_i(x^*).
\end{aligned}
$$

By Pinsker's inequality, the following inequality holds:

$$
\|y(x^*) - y(x^* \triangle \{i\})\|_\infty \leq \sqrt{\frac{1}{2} D_{\mathrm{KL}}(D_{x^*} \| D_{x^* \triangle \{i\}})}.
$$

Then, we have:

$$
\begin{aligned}
\sum_{i=1}^{n} \|y(x^*) - y(x^* \triangle \{i\})\|_\infty &\leq \sum_{i=1}^{n} \sqrt{\frac{1}{2} D_{\mathrm{KL}}(D_{x^*} \| D_{x^* \triangle \{i\}})} \\
&\leq \sqrt{n \sum_{i=1}^{n} \frac{1}{2} D_{\mathrm{KL}}(D_{x^*} \| D_{x^* \triangle \{i\}})} \\
&\leq \sqrt{\frac{\log 3}{4} \cdot \frac{a^2 T}{n} \epsilon^2}.
\end{aligned}
\tag{7}
$$

Therefore,

$$
\begin{aligned}
e_i(x^*) + e_i(x^* \triangle \{i\}) &= |y_i(x^*) - x_i^*| + |y_i(x^* \triangle \{i\}) - (1 - x_i^*)| \\
&\geq |y_i(x^*) - x_i^* - y_i(x^* \triangle \{i\}) + (1 - x_i^*)| \\
&\geq |1 - 2x_i^*| - |y_i(x^*) - y_i(x^* \triangle \{i\})| \\
&= 1 - |y_i(x^*) - y_i(x^* \triangle \{i\})|.
\end{aligned}
$$

Then, we obtain the following inequality by setting $\epsilon = \frac{1}{\sqrt{\log 3}} \cdot \frac{n^{\frac{3}{2}}}{a\sqrt{T}}$:

$$
\begin{aligned}
\sum_{i=1}^{n} \left( e_i(x^*) + e_i(x^* \triangle \{i\}) \right) &\geq \sum_{i=1}^{n} \left( 1 - |y_i(x^*) - y_i(x^* \triangle \{i\})| \right) \\
&\geq n - \sqrt{\frac{\log 3}{4} \cdot \frac{a^2 T}{n} \epsilon^2} \quad \text{(By equation 7)} \\
&= \frac{n}{2}.
\end{aligned}
\tag{8}
$$

Finally, if $S^*$ is chosen uniformly at random from $2^{[n]}$, $x^*$ also follows a uniform distribution (we denote $x^* \sim$ Unif). Then, we obtain the lower bound in Theorem 5:

$$
\begin{aligned}
\mathbb{E}[E_T] &= \mathbb{E}_{x^* \sim \text{Unif}}[E_T(x^*)] \\
&= \frac{\epsilon}{2n} \mathbb{E}_{x^* \sim \text{Unif}}\left[\sum_{i=1}^{n} e_i(x^*)\right] \\
&= \frac{\epsilon}{4n} \mathbb{E}_{x^* \sim \text{Unif}}\left[\sum_{i=1}^{n} \left(e_i(x^*) + e_i(x^* \triangle \{i\})\right)\right] \\
&\geq \frac{\epsilon}{4n} \mathbb{E}_{x^* \sim \text{Unif}}\left[\frac{n}{2}\right] \quad \text{(By equation 8)} \\
&= \frac{1}{8\sqrt{\log 3}} \frac{n^{\frac{3}{2}}}{a\sqrt{T}}.
\end{aligned}
$$

The second equality follows from the fact that if $x^*$ follows the uniform distribution, then $x^* \triangle \{i\}$ also follows the uniform distribution for any $i \in [n]$.

For algorithms without any restrictions, the following inequality holds instead of equation 6:

$$
\sum_{i=1}^{n} D_{\text{KL}}(D_{x^*} \| D_{x^* \triangle \{i\}}) \leq \frac{\log 3}{2} \cdot \frac{a^2 T}{n} \epsilon^2.
$$

Then, by setting $\epsilon = \frac{1}{\sqrt{\log 3}} \cdot \frac{n}{a\sqrt{T}}$, we have the same inequality as equation 8. Finally, we obtain the lower bound:

$$
\mathbb{E}[E_T] \geq \frac{1}{8\sqrt{\log 3}} \frac{n}{a\sqrt{T}}.
$$

$\square$

Next, we establish the lower bound for the sigmoid transfer function. The overall proof follows the same structure as in the linear case, with the differences stated as the following lemma.

**Lemma 1.** *Let $\rho(x) = \frac{2}{1+e^{-bx}} - 1$, and let the other definitions follow those used in the proof of Theorem 5. For any algorithm that satisfies Restriction 1, the KL divergence between two distributions $D_{x^*}$ and $D_{x^* \triangle \{i\}}$ satisfies the following inequality:*

$$
D_{\text{KL}}(D_{x^*} \| D_{x^* \triangle \{i\}}) \leq \frac{b^2 T}{4n^2} \epsilon^2.
$$

*For any algorithm without any restrictions, the following inequality holds:*

$$
D_{\text{KL}}(D_{x^*} \| D_{x^* \triangle \{i\}}) \leq \frac{b^2 T}{4n} \epsilon^2.
$$

*Proof.* By the *chain rule* for KL divergence, we have:

$$
\begin{aligned}
D_{\text{KL}}(D_{x^*} \| D_{x^* \triangle \{i\}}) &\leq D_{\text{KL}}((i_t, o_t)_{t=1}^{T-1}|_{x^*} \| (i_t, o_t)_{t=1}^{T-1}|_{x^* \triangle \{i\}}) \\
&\quad + \mathbb{E}_{(i_t, o_t)_{t=1}^{T-1}|_{x^*}}\left[\mathbb{1}\{i_T = i\} D_{\text{KL}}(\text{Ber}^{\pm}(\rho(\frac{\epsilon}{2n})) \| \text{Ber}^{\pm}(\rho(-\frac{\epsilon}{2n})))\right] \\
&= D_{\text{KL}}((i_t, o_t)_{t=1}^{T-1}|_{x^*} \| (i_t, o_t)_{t=1}^{T-1}|_{x^* \triangle \{i\}}) + \Pr\nolimits_{x^*}(i_T = i)(\frac{b\epsilon}{2n} \rho(\frac{\epsilon}{2n})) \\
&\qquad\qquad \text{(By the definition of KL divergence and } \rho(x)) \\
&\leq \sum_{t=1}^{T} \Pr\nolimits_{x^*}(i_t = i)(\frac{b\epsilon}{2n} \rho(\frac{\epsilon}{2n})).
\end{aligned}
$$

When the algorithm satisfies Restriction 1, it holds $\sum_{i=1}^{n} \sum_{t=1}^{T} \Pr_{x^*}(i_t = i) = T$. Therefore, we have the following inequality:

$$
\begin{aligned}
\sum_{i=1}^{n} D_{\text{KL}}(D_{x^*} \| D_{x^* \triangle \{i\}}) &\leq T \frac{b\epsilon}{2n} \rho(\frac{\epsilon}{2n}) \\
&\leq \frac{b^2 T}{4n^2} \epsilon^2 \quad (\because \rho(x) \leq bx \text{ for all } x > 0).
\end{aligned}
$$

Moreover, for algorithms without any restrictions, it follows $\sum_{i=1}^{n}\sum_{t=1}^{T}\text{Pr}_{x^*}(i_t = i) = nT$. Therefore, we obtain:

$$D_{\text{KL}}(D_{x^*}\|D_{x^*\triangle\{i\}}) \leq \frac{b^2 T}{4n}\epsilon^2.$$

$\square$

## B    NUMERICAL EXPERIMENT

### B.1    DEPENDENCE OF THE OPTIMIZATION ERROR ON THE PROBLEM PARAMETERS

In this section, we investigate, through numerical experiments, how the error achieved by the algorithm varies with the problem parameters ($T$ and $n$), and we verify that the empirical behavior is consistent with the theoretical analysis. We implemented the algorithm with linear and sigmoid transfer functions and investigated the dependence of the error on the number of oracle calls $T$ and the dependence of the error on the size of the ground set $n$. Fig.3, 4, 5, 6 display the empirical results together with the upper and lower bounds derived in this paper. As objective functions, we used nontrivial submodular functions derived from cut functions, so that neither the empty set nor the full set is necessarily a minimizer.

The experimental settings are as follows. The linear transfer parameter $a$ and the sigmoid transfer parameter $b$ were both set to 1. Algorithmic parameters are as specified in 1 and 2. For the experiments on the dependence of the error on the number of oracle calls $T$, we fixed the size of the ground set at $n = 5$ and ran the algorithms with $T \in \{500, 1000, 1500, 2000, 2500\}$. For the experiments on the dependence of the error on the size of the ground set $n$, we fixed the number of oracle calls at $T = 2000$ and varied $n \in \{4, 6, 8, 10, 12, 14\}$.

For each objective function, we ran the algorithm 100 times and recorded the average error between the algorithm's output and the optimal solution. This procedure was repeated for 100 randomly generated objective functions; the plots report the mean error across these functions together with the corresponding standard deviations.

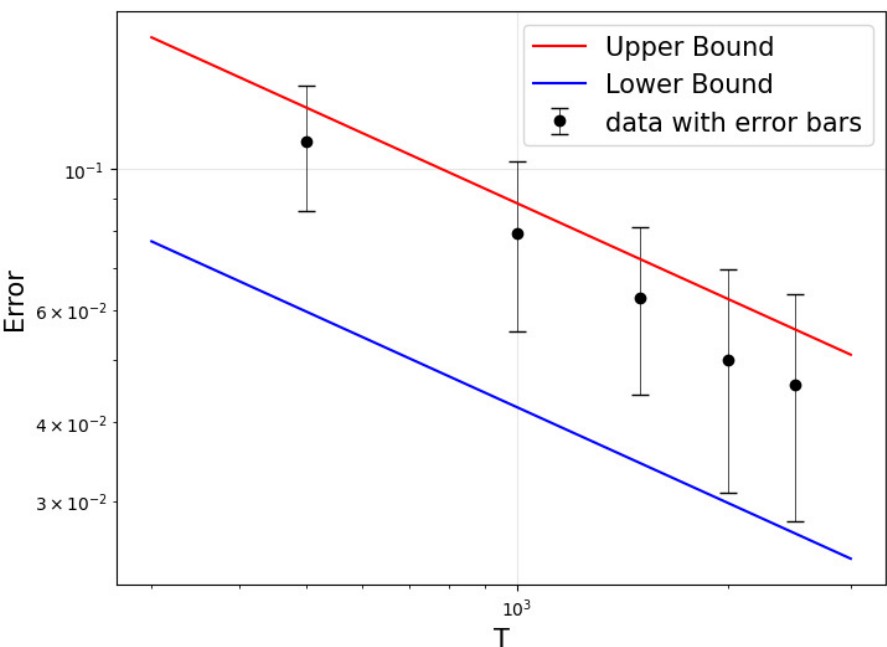

Figure 3: Dependence of the error on the number of oracle calls $T$ under the linear transfer function.

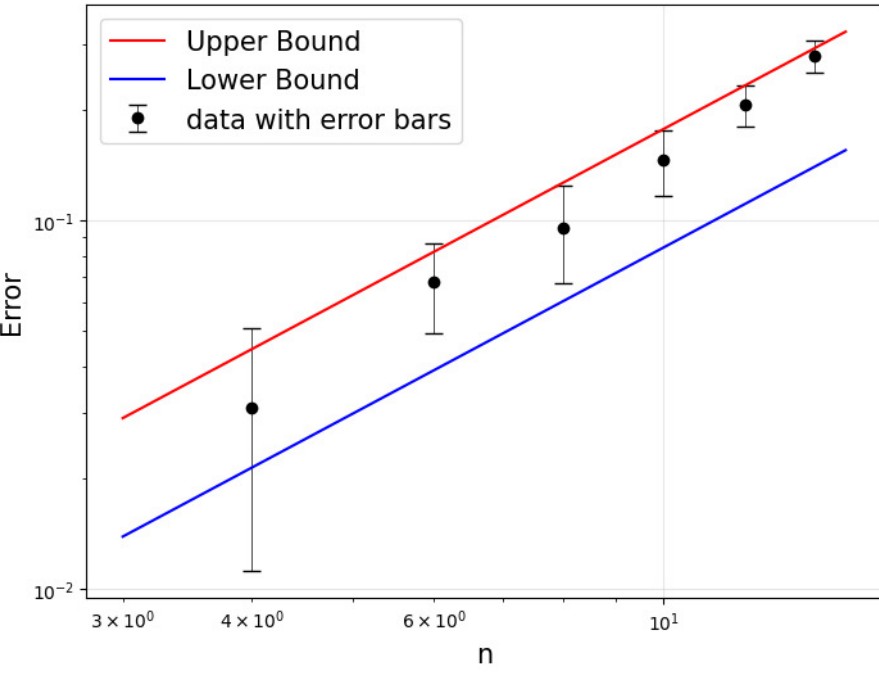

Figure 4: Dependence of the error on the size of the ground set $n$ under the linear transfer function.

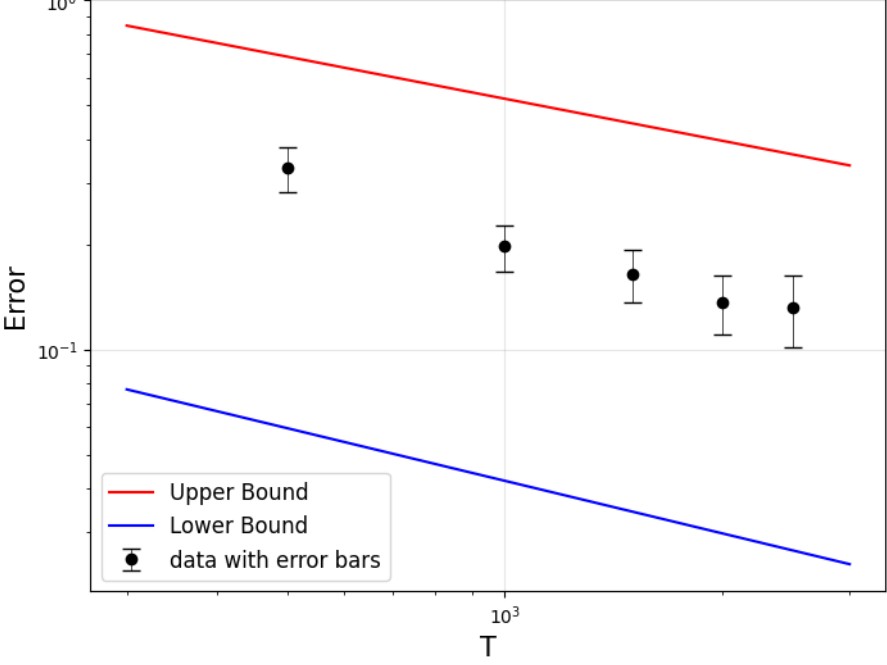

Figure 5: Dependence of the error on the number of oracle calls $T$ under the sigmoid transfer function.

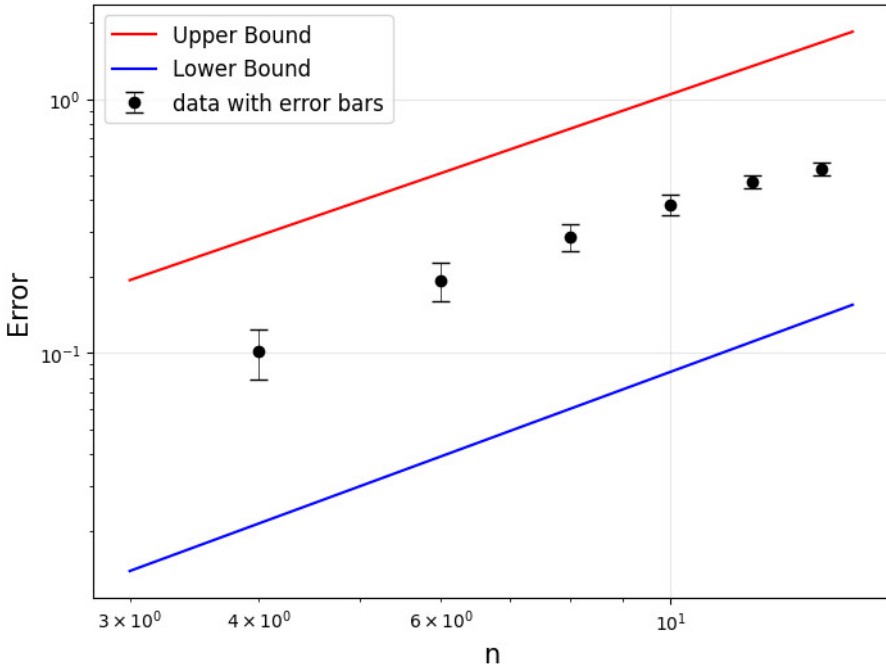

Figure 6: Dependence of the error on the size of the ground set $n$ under the sigmoid transfer function.

B.2 EXPERIMENTS UNDER MISSPECIFIED TRANSFER FUNCTIONS

In this section, we evaluate the performance of the proposed algorithm through numerical experiments in a misspecified transfer-function setting, i.e., when the transfer function assumed in the algorithm design does not match the actual transfer function that governs the behavior of the dueling oracle. Such misspecification commonly arises in practical applications, where there often exists a gap between the real-world problem and the mathematical model used for algorithm design; therefore, robustness to this type of misspecification is practically important. We also note that Section B.1 focused on the correctly specified setting, where the dueling oracle indeed follows the assumed linear or sigmoid transfer function.

In the experiments, we used the following two transfer functions: the clipped linear function $\rho_{\text{clip}}$ and the cubic function $\rho_{\text{cubic}}$ defined as

$$\rho_{\text{clip}}(x) = \min\{0.1, \max\{-0.1, x\}\},$$
$$\rho_{\text{cubic}}(x) = 2x - x^3.$$

Using the dueling oracle induced by each transfer function, we evaluate whether the algorithm designed for the linear transfer function (Algorithm 1) still achieves vanishing error. We set $n = 5$ and plot in Figure 7 how the error behaves as $T$ increases over $\{100, 200, 400, 800, 1600, 3200\}$. All other experimental settings are identical to those in Section B.1.

From Figure 7 we observe that the error approaches zero as $T$ increases for both transfer functions. Moreover, the convergence rate appears comparable to that in Section B.1, where the correct transfer function was used. This suggests that the proposed algorithm is reasonably robust to misspecification of the transfer function. In other words, the algorithm is expected to perform well even in more realistic scenarios where the comparison outcomes do not strictly satisfy the assumed model.

Although we are currently unable to provide a mathematically rigorous explanation for why the algorithm continues to perform well under such misspecified settings, the phenomenon can be intuitively understood through the interpretation discussed in the last paragraph of Section 3. Specifically, the transfer functions used in this section are not strictly linear, but they can be well approximated by a linear function in a neighborhood around zero. Therefore, when the difference in the function values used by the dueling oracle is sufficiently small (e.g., in regions close to the optimal solution), the

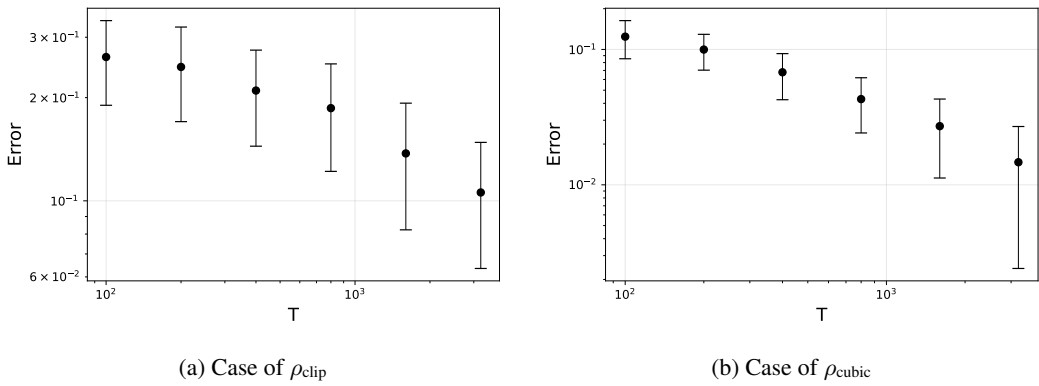

(a) Case of $\rho_{\text{clip}}$            (b) Case of $\rho_{\text{cubic}}$

Figure 7: Performance of Algorithm 1 under misspecified transfer functions.

behavior of the oracle becomes nearly indistinguishable from that of a linear transfer function. This explains why Algorithm 1, which assumes a linear transfer function, still performs reasonably well. Developing a theoretical framework to analyze and characterize this phenomenon is an interesting direction for future work.

