# OpenReview forum: "Submodular Function Minimization with Dueling Oracle"
_ICLR.cc/2026/Conference — ICLR 2026 Poster_

### Official Review · Reviewer_qsMZ · 2025-10-31

**Soundness:** 3
**Presentation:** 3
**Contribution:** 2
**Rating:** 4
**Confidence:** 3

**Summary:**

This paper studies the problem of submodular function minimization under the noisy dueling oracle setting, where the algorithm receives noisy pairwise comparisons between subsets rather than direct function evaluations, and the oracle's response is further determined by a transfer function. The authors establish both upper and lower bounds on the achievable error rate. In particular, they proposed algorithms that achieve an error rate of $O(n^{3/2}/\sqrt{T})$ and $O(n^{7/5}/{T}^{2/5})$, respectively for linear and sigmoid transfer functions. In particular, the achieved error rate under the case of a linear transfer function matches the result of the lower bound.

**Strengths:**

1. The problem of submodular optimization under the i.i.d. bandit feedback setting has been extensively studied in prior work and has broad applications across various machine learning tasks. In this regard, the paper addresses an important and timely problem of clear relevance to the ICLR community.

2. The authors present a thorough theoretical analysis, deriving both upper and lower bounds on the error rate for the considered setting.

3. The proposed algorithm attains the optimal regret bound in the special case of a linear transfer function.

**Weaknesses:**

1. The problem of submodular optimization under the i.i.d. bandit feedback setting has been studied in many previous works. Compared with existing works, the key assumption is the existence of the transfer function. While the authors provide several motivating examples related to submodular optimization with noisy dueling bandit feedback, it remains unclear how the proposed transfer function appropriately models those scenarios.

 2. The paper does not clearly articulate its technical contributions. It would be helpful for the authors to explicitly emphasize the novel aspects of their analysis. In particular, highlighting the key distinctions from prior work and elaborating on the specific technical challenges addressed would make the theoretical contributions more compelling and substantial.

**Questions:**

1. Compared to existing work, the main distinguishing feature is the transfer function. Do these assumptions introduce specific technical challenges that must be addressed in the analysis? What are the novel elements in the proofs that arise from these assumptions?

2. For the sigmoid and linear transfer functions, could the authors clarify why these assumptions are reasonable and important in the examples provided? In particular, the linear transfer function appears somewhat impractical, and additional explanation regarding its relevance would be helpful.

---

> ### Author Response · Authors · 2025-11-20
>
> Thank you very much for taking the time to review our paper. We also appreciate your insightful comments, questions, and valuable feedback. We hope that the responses below adequately address your concerns and questions.
>
>
>
> > The paper does not clearly articulate its technical contributions. It would be helpful for the authors to explicitly emphasize the novel aspects of their analysis. In particular, highlighting the key distinctions from prior work and elaborating on the specific technical challenges addressed would make the theoretical contributions more compelling and substantial.
>
> The technical contributions, challenges, and novelties are briefly described in lines 110–155 of the revised manuscript. In response to the reviewers’ comments, we have added more detailed explanations of the difficulties involved in designing the algorithm and proving the upper bound under the sigmoid transfer function, as well as the challenges that arise in the lower-bound analysis.
>
> > Compared to existing work, the main distinguishing feature is the transfer function. Do these assumptions introduce specific technical challenges that must be addressed in the analysis? What are the novel elements in the proofs that arise from these assumptions?
>
> Your understanding that the presence of the transfer function is central to both the novelty of the problem formulation and the main technical challenges in our work is entirely correct. In particular, the major technical difficulties arise when dealing with the nonlinear sigmoid transfer function, as explained in lines 114–137 of the revised manuscript. In this case, it becomes difficult to construct an unbiased gradient estimator of the kind commonly used in standard bandit-feedback settings.
>
> To address this challenge, we employ a gradient estimator based on Firth’s method. This introduces a bias term that must be carefully accounted for in the error analysis, which makes the proof substantially different from prior work and requires additional nontrivial ideas for both the analysis and the parameter tuning.
>
> > The problem of submodular optimization under the i.i.d. bandit feedback setting has been studied in many previous works. Compared with existing works, the key assumption is the existence of the transfer function. While the authors provide several motivating examples related to submodular optimization with noisy dueling bandit feedback, it remains unclear how the proposed transfer function appropriately models those scenarios.
>
> > For the sigmoid and linear transfer functions, could the authors clarify why these assumptions are reasonable and important in the examples provided? In particular, the linear transfer function appears somewhat impractical, and additional explanation regarding its relevance would be helpful.
>
>
> The reasons why each transfer function is particularly important are explained in the two paragraphs following their introduction, in lines 317–342 of the revised manuscript. In addition, we have included experimental results for cases where the transfer function differs from both of these forms in Section B.2 (“Experiments under misspecified transfer functions”).
>
> We believe that the linear transfer function is important as any smooth nonlinear transfer function can be locally approximated by a linear function. As described in the last paragraph of Section 3 and in the final paragraph of Section B.2, if the transfer function is smooth around 0 and the objective-function values of the queried pair are not far apart, the feedback obtained from the oracle can be well approximated by that of the linear transfer-function case. Indeed, the experimental results in Section B.2 suggest that even when the true transfer function is nonlinear, the algorithm assuming a linear transfer function (Algorithm 1) can still perform well. Although we do not yet have a theoretical explanation for this phenomenon, we believe it is a practically useful heuristic.
>
> We have also added several references to motivate the importance of considering the sigmoid transfer function.

---

### Official Review · Reviewer_eqdV · 2025-11-01

**Soundness:** 3
**Presentation:** 4
**Contribution:** 3
**Rating:** 8
**Confidence:** 2

**Summary:**

This work studies submodular function maximization under a new setting where the algorithm does not observe the function values, but only pairwise noisy preferences returned by an oracle. This paper formalizes how duelling responses depend on difference function values through a transfer function (linear, sigmoid etc). For the linear case, the authors propose an SGD-based algorithm using Lvasz extension and show provable optimality, For the sigmoid case, the authors use Firth's bias-correction technique to obtain a theoretical guarantee.

**Strengths:**

The introduction of the problem.
The hardness of theoretical results is also provided the algorithms are provably optimal. Experimental results give insights are how errors depend on n and T

**Weaknesses:**

The algorithms heavily rely on continuous extensions. Thus,a concern is overall efficiency of the algorithms could be limited compared to the discrete methods. Is it possible to design algorithms without using the continuous extensions?

The analysis largely adopts techniques from earlier works. How do proof techniques differ>

Presentation could be improved: Adding brief description showing why Algorithm 1 naturally satisfies Restriction 1 could be helpful.

**Questions:**

Please see the weakness.

---

> ### Author Response · Authors · 2025-11-20
>
> Thank you very much for taking the time to review our paper, and for providing many valuable comments and suggestions. We hope that the responses below adequately address your questions and concerns.
>
> > The algorithms heavily rely on continuous extensions. Thus, a concern is overall efficiency of the algorithms could be limited compared to the discrete methods. Is it possible to design algorithms without using the continuous extensions?
>
> At present, we have not been able to identify an efficient approach that avoids the use of continuous extensions. For the related problem of online submodular minimization, Hazan \& Kale (2009) have proposed an FTPL-based method that does not rely on continuous extensions in their Section 3.1; however, both in terms of computational efficiency and error (regret) guarantees, approaches based on continuous extensions in Section 3.2 appear to perform better in that setting.
>
> That said, these results do not rule out the possibility of developing efficient methods that do not rely on continuous extensions. We believe that exploring this direction remains an interesting avenue for future research. Accordingly, we have added a brief discussion of this possibility to the conclusion section of the revised manuscript.
>
> > The analysis largely adopts techniques from earlier works. How do proof techniques differ>
>
> The technical challenges and novel aspects of our work are described in lines 114–148 of the revised manuscript. In response to the reviewers’ comments, we have added a more detailed explanation of the difficulties involved in designing the algorithm and proving the upper bound under the sigmoid transfer function, as well as the challenges that arise in the lower-bound analysis.
>
> For the upper-bound proof in the case of the sigmoid transfer function, several challenges arise due to the fact that, unlike in the standard SFM-based approaches, we cannot use an unbiased gradient estimator. To address this issue, we employ a gradient estimator based on Firth’s method. This introduces a bias term that must be carefully accounted for in the error analysis, which makes the proof more intricate than in prior work and requires additional nontrivial ideas for both the analysis and parameter tuning.
>
> > Presentation could be improved: Adding brief description showing why Algorithm 1 naturally satisfies Restriction 1 could be helpful.
>
> We fully agree with your point. We have added an explanation immediately after Restriction 1. We sincerely appreciate this very helpful suggestion.

---

### Official Review · Reviewer_BQte · 2025-11-03

**Soundness:** 4
**Presentation:** 4
**Contribution:** 3
**Rating:** 6
**Confidence:** 3

**Summary:**

This work studies submodular function minimization (SMF) with a *dueling oracle* (i.e., an oracle that takes two sets $S$ and $S'$, and
probabilistically returns which set has the higher value with probability $\frac{1}{2} + \frac{1}{2} \rho( f(S) - f(S'))$ for
a fixed *transfer function* $\rho : [-1, 1] \rightarrow [-1, 1]$. A dueling oracle has been used for multi-armed bandit problems
in the context of convex optimization (e.g., [Saha et al., ICML 2021] and [Saha et al., ICML 2025]), but this work is the first to study them for SMF.
The authors give strong upper and lower bounds for the following transfer functions: linear, sigmoid, and general (see Table 1).
Overall, this is a nice and very well written theoretical paper.

**Strengths:**

- Good classification of upper and lower bounds for SMF for different transfer functions in Table 1.
- Combines a nice set of tools: Lovasz extension (continuous methods), Yao's principle for analysis
- Very well written introduction with motivating applied ML examples in Example 1 and Example 2

**Weaknesses:**

- Lack of experiments
- Submodular function minimization isn't the strongest fit for ICLR, but there is precidence

**Questions:**

- [264] How does the randomness of the dueling oracle work? Does it give the
  same noisy response for any given pair of subsets $S$ and $S'$, or are there
  independent coin flips each time the oracle is called?

---

> ### Author Response · Authors · 2025-11-20
>
> Thank you very much for taking the time to read our paper and for providing valuable feedback. We hope that our responses below adequately address your concerns.
>
> > Lack of experiments
>
> The numerical experiments using synthetic data are presented in Section B of the Appendix. Section B.1 was included in the original submission, and in this revision we have added Section B.2. In Section B.2, to better approximate realistic scenarios, we conduct experiments under a misspecified transfer-function setting, where the observed data do not fully satisfy the assumed model, and we report the corresponding results. As shown in Figure 7 of this section, the proposed algorithm continues to perform well even under such misspecification, which suggests that it is robust to errors in the transfer-function model.
>
> > [264] How does the randomness of the dueling oracle work? Does it give the same noisy response for any given pair of subsets $S$ and $S'$, or are there independent coin flips each time the oracle is called?
>
> Thank you very much for pointing out this important issue. Your interpretation—the latter one, where each output is an independent coin flip—is correct. To clarify this point, we have added the following sentences to lines 275–277 of the revised manuscript:
>
> > *Note that, as in prior works on dueling bandits and dueling convex optimization (Saha et al., 2021a;b), all outputs from the dueling oracle are assumed to be independent. Therefore, even if the same pair is queried multiple times, the outputs are not necessarily identical.*
>
> This is a crucial assumption that should indeed have been explicitly stated, and we sincerely appreciate your bringing this to our attention.

---

### Official Review · Reviewer_2Y67 · 2025-11-10

**Soundness:** 3
**Presentation:** 3
**Contribution:** 2
**Rating:** 6
**Confidence:** 3

**Summary:**

**Main contribution:**
This paper presents the first study of submodular function minimization (SFM) using only a dueling oracle, which provides noisy pairwise comparisons rather than exact function values. The authors propose and analyze algorithms for this setting, establishing error bounds for both linear and sigmoid oracle transfer functions.

**Problem formulation:**
The objective is to minimize a submodular function $f:2^{[n]}\rightarrow[0,1]$, which is a set function satisfying $f(X)+f(Y)\ge f(X\cup Y)+f(X\cap Y)$ for all $X, Y\subseteq[n]$. Access to the function is restricted to a dueling oracle, which, when queried with two sets $(S,S^{\prime})$, returns a random binary response $o\in\{\pm1\}$. The probability of the response is governed by a transfer function $\rho$ based on the value difference, e.g., $Pr(o=+1)=\frac{1}{2}+\frac{1}{2}\rho(f(S)-f(S^{\prime}))$. The goal is to minimize the additive error $E_{T}:=f(\hat{S})-min_{S\in2^{[n]}}f(S)$ after $T$ oracle calls.

**Main results**
For a linear transfer function, the paper provides an algorithm with an error rate of $O(n^{\frac{3}{2}}/\sqrt{T})$, which is proven to be optimal for constrained algorithms by a matching lower bound. For a sigmoid transfer function, the authors design an algorithm with an error rate of $O(n^{\frac{7}{5}}/T^{\frac{2}{5}})$.

**Technique/algorithm summary:**
The algorithms are based on applying Stochastic Gradient Descent (SGD) to the Lovász extention, which is a continuous convex relaxation of the submodular function. In the linear case, an unbiased subgradient estimator for SGD is constructed directly from the dueling oracle's response. Because unbiased estimation is infeasible for the sigmoid function, that algorithm instead employs Firth's method (Firth, 1993) to create a low-bias estimator from the logistic regression model, thereby mitigating error accumulation.

**Experiment summar **
The paper includes numerical experiments that implement the proposed algorithms for both linear and sigmoid transfer functions on submodular cut functions. The result empirically validate the theoretical findings by plotting the error's dependence on the number of oracle calls ($T$) and the ground set size ($n$), showing the data falls between the derived upper and lower bounds.

**Strengths:**

The theoretical results are solid.

**Weaknesses:**

Even though this is mainly a theoretical paper, the paper also provides some experimental supports. However, it would be good to provide experiments on some real world data than just numerical experiments.

**Questions:**

.

**Details Of Ethics Concerns:**

.

---

> ### Author Response · Authors · 2025-11-20
>
> Thank you very much for taking the time to review our paper and for providing valuable feedback.
>
> > However, it would be good to provide experiments on some real world data than just numerical experiments.
>
> We fully agree with your suggestion that evaluating the practical usefulness of the method on real-world data would be highly valuable. Unfortunately, within the available time frame, we were not able to prepare experiments using real-world datasets. Instead, to better approximate realistic deployment scenarios, we conducted experiments under a misspecified transfer-function setting, where the observed data do not fully satisfy the assumed model. We have added these results to Section B.2 of the Appendix. As shown in Figure 7 of that section, the proposed algorithm continues to perform well even under such misspecification.

---

### Author Response · Authors · 2025-11-20
**Revision summary**

We would like to express our sincere gratitude to all reviewers for their careful evaluations and many valuable comments. Based on your feedback, we have made the following revisions. Except for minor typographical fixes, all modifications and additions are highlighted in blue.

* **p.3:** Added an explanation of the technical challenges and the novelty of our analysis.
* **Section 3, first paragraph:** Clarified that all outputs from the dueling oracle are independent.
* **Section 3, last two paragraphs:** Updated the discussion on why the sigmoid and linear transfer functions play an important role.
* **Immediately after Restriction 1 in Section 6:** Added a statement that the algorithms satisfy Restriction 1.
* **Section 7:** Added a discussion, as future work, on approaches that do not rely on continuous extensions.
* **Appendix Section B.2 and Figure 7:** Added a new section presenting numerical experiments for misspecified transfer functions. We numerically evaluate how Algorithm 1 behaves when the actual transfer function is neither linear nor sigmoid.

We believe that the manuscript has been significantly improved thanks to your insightful comments. We hope that the revisions and our responses adequately address your concerns and questions.

---

### Meta-Review · Area_Chair_w7e3 · 2026-01-15

**Summary:**

Paper studies submodular function minimization with only noisy pairwise (dueling) feedback, formalized via transfer functions (linear, sigmoid). Main contributions are tight upper/lower bounds (optimal up to constants in the linear/constrained regime) and a sigmoid-case algorithm using Firth bias correction to handle lack of unbiased gradients; experiments are synthetic but include a misspecified transfer check.

**Reviewer Concerns:**

Addressed: independence of oracle draws clarified; novelty/technical challenges (esp. sigmoid) better explained; Restriction 1 satisfaction clarified; added misspecification experiments; acknowledged “no real-world experiments” gap.

Still outstanding: limited empirical validation (no real datasets/applications), and one reviewer remains unsure the transfer-function modeling matches motivating scenarios (linear case realism), though authors give a reasonable “local linearization” argument + robustness evidence.

**Reviewer Scores:**

eqdV: likely stays 8.
2Y67: likely stays 6.
BQte: likely stays 6.
qsMZ: could move 4 to 5

---

### Decision · Program_Chairs · 2026-01-26

Accept (Poster)